

# Laboratory characterisation of the radiation temperature error of radiosondes and its application to the GRUAN data processing for the Vaisala RS41

Christoph von Rohden[1], Michael Sommer[1], Tatjana Naebert[1], Vasyl Motuz[2], and Ruud J. Dirksen[1]

[1]GRUAN Lead Centre, Deutscher Wetterdienst, Meteorologisches Observatorium Lindenberg, Am Observatorium 12, 15848 Tauche/Lindenberg, Germany
[2]Department of Aerodynamics and Fluid Mechanics, Brandenburg University of Technology, Cottbus-Senftenberg, Siemens-Halske-Ring 15a, 03046 Cottbus, Germany

**Correspondence:** Christoph von Rohden (Christoph.Rohden-von@dwd.de)

**Abstract.**

   The paper presents the Simulator for Investigation of Solar Temperature Error of Radiosondes (SISTER), a setup that was developed to quantify the solar heating of the temperature sensor of radiosondes under laboratory conditions by recreating as closely as possible the atmospheric and illumination conditions that are encountered during a daytime radiosounding ascent.

SISTER controls the pressure (3 hPa to 1020 hPa) and ventilation speed of the air inside the windtunnel-like setup to simulate the conditions between the surface and 35 km altitude, to determine the dependence of the radiation temperature error on the irradiance and the convective cooling. The radiosonde is mounted inside a quartz tube, while the complete sensor boom is illuminated by an external light source to include the conductive heat transfer between sensor and boom. A special feature of SISTER is that the radiosonde is rotated around its axis to imitate the spinning of the radiosonde in flight. The characterisation

of the radiation temperature error is performed for various pressures, ventilation speeds and illumination angles, yielding a 2D-parameterisation of the radiation error for each illumination angle, with an uncertainty smaller than 0.2 K ($k = 2$) for typical ascend speeds. This parameterisation is applied in the GRUAN processing for radiosonde data, which relies on the extensive characterisation of the sensor properties to produce a traceable reference data product which is free of manufacturer dependent effects. The GRUAN radiation correction model combines the laboratory characterisation with model calculations

of the actual radiation field during the sounding to estimate the correction profile. In the second part of this paper it is described how this procedure was applied in the development of the GRUAN data product for the Vaisala RS41 radiosonde (version 1, RS41-GDP.1). The magnitude of the averaged correction profile increases gradually from 0.1 K at the surface to approximately 0.8 K at 35 km altitude. Comparison between sounding data ($N = 154$) that were GRUAN-processed and Vaisala-processed reveal that the daytime differences are smaller than $+0.1$ K (GRUAN $-$ Vaisala) in the troposphere and increase above the

tropopause steadily with altitude to $+0.35$ K (GRUAN $-$ Vaisala) at 35 km. These differences are just within the limits of the combined uncertainties (with coverage factor $k = 2$) of both data products, meaning that the GRUAN processing and the Vaisala processing are in agreement.



# 1 Introduction

For almost a century, radiosondes have been successfully providing essential measurements of the state of the atmosphere at unmatched vertical resolution up to an altitude of approximately 35 km. Nowadays, more than 800 radiosoundings are performed each day by the global observational network, providing essential measurement data to, for example, numerical weather prediction (NWP). These long-term global data records have great value for climate monitoring (Elliott and Gaffen, 1991; Gaffen, 1994; Seidel et al., 2009). However they often contain inhomogeneities caused by changes in measurement

systems (Elliott and Gaffen, 1991), and uncorrected measurement errors as well as undisclosed data processing of commercial radiosondes affect the quality of the measurement data. The need for high-quality reference data motivated the founding of the GCOS Reference Upper Air Network (GRUAN) (www.gruan.org) in 2008 (Seidel et al., 2009; Bodeker et al., 2016).

One of the main goals of the GRUAN is to perform reference observations of profiles of Essential Climate Variables (ECVs) such as atmospheric temperature and humidity, for the purpose of monitoring climate change, and for other applications such

as NWP and satellite validation. Essential criteria for establishing a reference observation include measurement-traceability, correction of all known errors and biases, and the availability of measurement uncertainties. Manufacturer-processed data often do not fulfil the criteria for a reference data product, mainly because of the application of undisclosed correction algorithms in the data processing, inadequate correction of measurement errors, or missing measurement uncertainties. In contrast, it is a prerequisite for GRUAN data products (GDPs) to comply with the criteria for a reference product, for which the applied

correction methods are based on extensive characterisation of the instrument and its sensors. As a consequence, the development of GRUAN data processing for a specific instrument requires considerable effort and can be a time-consuming process. Currently, GDPs are available for the Vaisala RS92 (Dirksen et al., 2014) and Meisei RS-11G (Kobayashi et al., 2019; Kizu et al., 2018) radiosonde models, and for Global navigation satellite system precipitable water vapor (GNSS-PW), while data products for additional radiosonde models, as well as for other measurement techniques such as lidar or Microwave radiometer

(MWR) are under development.

The main error source for daytime radiosonde temperature measurements is the warming of the temperature sensor by solar radiation, a problem that has affected temperature measurements since the beginning of balloon-borne aerological observations. A relatively easy way to estimate the radiation temperature error uses the observed day-night difference, which for previous radiosonde models was shown to amount up to 2 K in the stratosphere (McInturff et al., 1979; Luers, 1990). The temperature

sensor is usually located at the far end of a sensor boom, to minimise contamination by air flowing over the casing of the radiosonde. The temperature measurement by the radiosonde is a contact measurement, where the sensor assumes the temperature of the surrounding air, with heat transfer between both media acting to reduce the differences. Absorption of solar radiation by the temperature sensor heats the sensor, which is counter-acted by convective cooling, conduction and radiation, where convective cooling is considered the dominant factor (Luers, 1990). The efficiency of the convective cooling reduces





with the decreasing pressure at higher altitudes, leading to an increase of the temperature error with altitude, which for current
radiosonde types amounts to approximately 1 K at 30 km altitude.

The exact quantification, and correction, of the radiation temperature error is a complicated and challenging effort, which is
exacerbated by the fact that there is no instrument available for in situ reference measurements of the air temperature that would
allow for an independent comparison of the radiosonde temperature measurements. Over time, different methods have been

used to estimate the radiation temperature error. A theoretical approach employed by, e.g. McMillin et al. (1992); Luers and
Eskridge (1995); Luers (1997) involves solving the heat balance for the irradiated temperature sensor, which requires detailed
knowledge of the thermodynamic and radiative properties of the sensor boom components, as well as thorough evaluation of
the aerodynamics of the air flow around the sensor boom. The accuracy of this approach is limited by the assumptions and
approximations that are necessary for material properties and for the modelling of the pressure-dependent convective heat

exchange.

A widely used method is the comparison with observations from other measurement techniques, such as space borne re-
mote sensing instruments. Examples of this are the work by Haimberger et al. (2008) to estimate warm-biases in long-term
radiosonde upper-air records relative to Microwave Sounding Unit (MSU) radiances, or the comparison with collocated GPS
Radio Occultation (GPS-RO) observations by Sun et al. (2013) and Ho et al. (2017). Such comparisons provide a statistical

estimate of the temperature bias between the radiosonde and the satellite instruments in question, but conclusive findings are
usually not possible because satellite-based temperature profiles do not constitute a calibrated reference. Other limitations that
complicate the comparison are for example that the timing of the radiosoundings needs to be adjusted to match satellite over-
passes to reduce the collocation error, and that accurate observations by microwave and infrared remote sensing instruments
require cloud free scenes.

A few attempts have been made to determine the radiation temperature error by a direct comparison of in situ instruments
(e.g. Schmidlin et al., 1986). This limited number of studies is mainly caused by the lack of a true, independent reference
for in situ temperature observations. Philipona et al. (2013) derived a correction from an in-flight experiment, where the solar
radiation temperature error was derived using the temperature difference between two identical thermocouple sensors, one
exposed and the other shielded from sun light, while simultaneously recording the shortwave and longwave fluxes. The resulting

altitude-dependent correction amounts to approximately 1 K at 35 km. Using a similar approach, Lee et al. (2018b) estimate
the temperature error from the bias between differently coated thermistors on the same radiosonde, which is subsequently used
to correct the measured temperature profile. The estimated uncertainty for this correction is 0.49 K ($k = 1$).

Following GRUAN methodology, the characterisation of the temperature sensor is performed in a traceable laboratory envi-
ronment, with a setup that can simulate the conditions that occur in flight. The result of this sensor characterisation is the basis

for the correction algorithms that are employed in the GRUAN data processing that is addressed in this paper. The warming
of the temperature sensor by solar radiation is determined as a function of pressure, flux, ventilation speed, and illumination
angle. The actinic flux is needed to relate the laboratory characterisation of the sensor's susceptibility to radiation to the actual
temperature error. By lack of information on the actual solar radiation profile in the GRUAN correction, the actinic flux is esti-
mated with a radiative transfer model (RTM), which calculates direct and diffuse upward and downward radiance components





while accounting for the changes of the location of the radiosonde and of the solar position during the ascent. The correction algorithm combines both components to estimate the temperature error for each data point of the ascent. Luers and Eskridge (1995) showed that longwave heating of a sensor with a metallic coating is much smaller than the heating by shortwave (visible) radiation, so that the GRUAN correction only considers the effects from shortwave radiation.

This approach was first applied for the GRUAN data processing of the Vaisala RS92 radiosonde, where the characterisation of the radiation temperature error was performed with the radiosonde inserted in a vacuum chamber, using the Sun as a light source (Dirksen et al., 2014). This chamber generated an internal airflow that is comparable to the ventilation experienced during a radiosonde ascent, at pressures between ambient and 3 hPa. Another, similar, setup was used in the development of the data processing for the Meisei RS-11G (Kizu et al., 2018), and recently Lee et al. (2020) built a setup to investigate the radiation temperature error at temperatures that prevail in the stratosphere, but so far the latter setup was not applied for GDP development. Several restrictions concerning the orientation of the sensor with regard to the light source and the air flow limited the ability of the chamber used by Dirksen et al. (2014) to realistically render the in-flight conditions, and this prompted the construction of the custom-build Simulator for Investigation of Solar Temperature Error of Radiosondes (SISTER) setup at Lindenberg observatory, which is described in this paper. SISTER is an air tight wind tunnel that produces ventilation speeds up to 7 m s$^{-1}$ at pressures between ambient and 3 hPa. Notable features of the SISTER are that the wind tunnel is wide enough to enable the axial rotation of the radiosonde with unfolded sensor boom, which allows to recreate the continuously changing illumination conditions resulting from the spinning of the radiosonde in flight. Furthermore, the temperature sensor together with a sizeable part of the sensor boom are illuminated. This approximates the daytime situation where the sensor boom is continuously exposed to sunlight, and this includes the influence of the sensor boom in the determination of the radiation error. Temperature sensors are usually kept small, as the radiation error scales with sensor size (de Podesta et al., 2018), but conductive heating from the energy absorbed by the comparably large area of the sensor boom can be a relevant factor. In the characterisation of the RS92, Dirksen et al. (2014) found indications that the sensor boom contributes considerably to the overall heating of the temperature sensor. This is supported by Computational Fluid Dynamics (CFD) model calculations by Han et al. (2018), that showed that the heating of a thin platinum-wire sensor is mainly caused by the conductive heating from the illuminated circuit board instead of the solar irradiation of the sensor wire. Finally, with SISTER the illumination geometry of the radiosonde can be adjusted to simulate solar positions that are representative for tropical, mid-latitude and arctic regions.

SISTER is used for the development of the GDP for the Vaisala RS41 radiosonde. Currently, 23 GRUAN sites employ the RS41 as operational radiosonde, which effectively makes it the backbone of GRUAN in terms of upper air soundings and which illustrates the importance to develop a GDP for the RS41. Dirksen et al. (2020) discuss how the replacement of the RS92 by the RS41 poses a special challenge for GRUAN as a reference network, and the strategy that is adopted to reduce inhomogeneities in the GRUAN data record.

The structure of this paper is as follows: Sect. 2 describes the SISTER setup including the LDA-based characterisation of the flow pattern, Sect. 3 describes the measurements to characterise the radiation temperature error, Sect. 4 describes the analysis of these measurements and the resulting parameterisation of the susceptibility of the RS41 temperature sensor to solar radiation, Sect. 5 describes the RTM simulations to calculate the actinic flux, Sect. 6 describes the actual correction





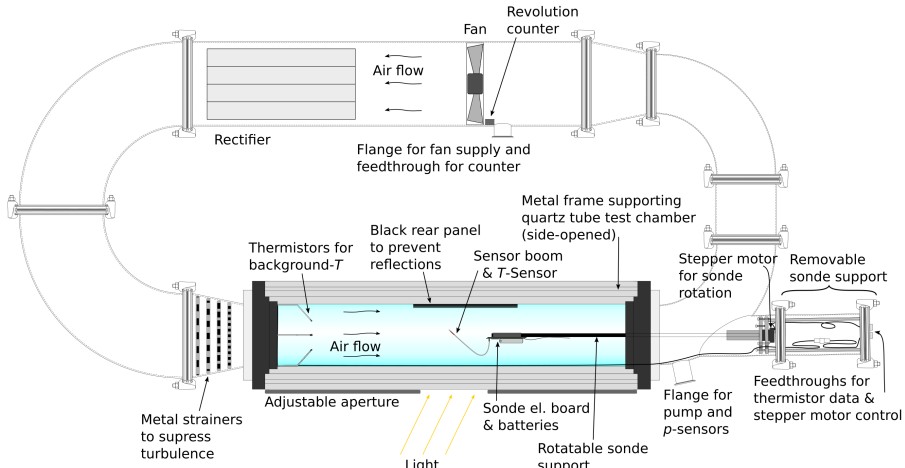

**Figure 1.** Schematic diagram of the SISTER radiation test chamber (top view).

algorithm that is applied in the GRUAN data processing, Sect. 7 presents a comparison between the GRUAN-processing and the Vaisala-processing of the RS41's temperature profiles, and Sect. 8 provides a summary and outlook.

## 2 Radiation test facility

### 2.1 Design considerations

SISTER (Simulator for Investigation of Solar Temperature Error of Radiosondes) is designed to simulate as closely as possible the conditions that are encountered during a typical radiosounding ascent. This involves controlling parameters such as pressure, ventilation speed and the illumination geometry. The more challenging requirements for rendering realistic illumination conditions involve the illumination of the temperature sensor together with a considerable part of the neighbouring sensor boom, as well as at the same time the continuous rotation of the sensor boom around the longitudinal axis to simulate the radiosonde's spinning in flight. In addition, a stable airflow of approximately $5\,\mathrm{m\,s^{-1}}$ at pressures between $1000\,\mathrm{hPa}$ and $3\,\mathrm{hPa}$ is required to mimic the ventilation by ambient air of the radiosonde during ascent. These requirements resulted in the design of an air-tight wind tunnel-like construction that is wide enough to hold a rotating radiosonde with unfolded sensor boom, and that can be operated between ambient and low pressure.

The radiation test facility, shown in Figs. 1 and 2, is a rectangular-shaped wind tunnel with external dimensions of approximately $1\,\mathrm{m} \times 2\,\mathrm{m}$. It is assembled of stainless steel tubes of $153\,\mathrm{mm}$ and $213\,\mathrm{mm}$ diameter. The actual measurement chamber, located in one of the middle legs, is formed by a $180\,\mathrm{mm}$ wide, $1\,\mathrm{m}$ long quartz tube in which the radiosonde is mounted. A membrane vacuum pump is used to set the pressure. The circulation of the air inside the chamber is driven by a fan that is located in the leg opposite to the measurement chamber. The fan's diameter is comparable to that of the metal tube in which it is mounted, to generate a radially-uniform flow, and a rectifier behind the fan serves to suppress turbulence. The radiosonde is





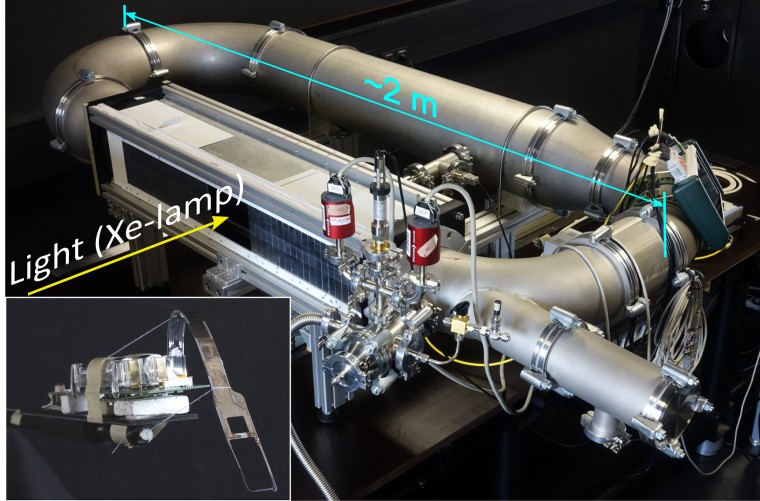

**Figure 2.** Photograph of SISTER. The inset shows how the radiosonde electronic board and sensor boom are mounted on the sonde holder.

mounted on a rod along the longitudinal axis of the quartz tube. To imitate the spinning of the radiosonde during ascent, the
rod is rotated at constant speed by a stepper motor. The rotation period can be 1 s or longer, and fixed positions can be selected
as well.

The sensor boom of the radiosonde is illuminated by a 2500 W Xe arc light source, generating a collimated beam with a
divergence of a few degrees. A manual aperture limits the beam diameter to approximately 20 cm at the centre of the quartz
tube. This is wide enough to illuminate the entire cross section of the quartz tube, so that for any orientation of the radiosonde,
the temperature sensor and a large part of the sensor boom are illuminated. Prior to the measurements, the radiosonde's casing
is removed to reduce its cross-section and thus to minimise the disturbance of the air flow, leaving only the electronic board
with batteries and the sensor boom. The sensor boom is bent at an angle of 45° with respect to the flow direction, and kept
in place using thin threads (see Fig. 1 and the inset in Fig. 2). The illumination angle, which represents the solar elevation
angle, can be varied by turning the entire chamber around an (imaginary) vertical axis that runs through the position of the
radiosonde's temperature sensor, and the flux is varied by changing the distance between the chamber and the light source.

## 2.2   Ventilation speed

The airflow at the position of the radiosonde is characterised using 2D-Laser Doppler Anemometry (LDA, e.g. Albrecht et al.,
2003). This optical method of flow velocity measurement is non-invasive, and in a closed volume under low-pressure conditions
it is the only method that does not require additional designs, such as a sealed sluice, for the insertion of a measuring probe.
Along with high spatial and temporal resolution, another advantage of LDA is that no calibration is required. This is particularly
important for an setup such as SISTER, where measurements are performed at different pressures.

LDA measurements rely on seeding with particles, that are carried along in the air flow. The measurement principle is
the detection of the Doppler shift of the laser light scattered off the moving aerosol particles. The aerosols are produced in



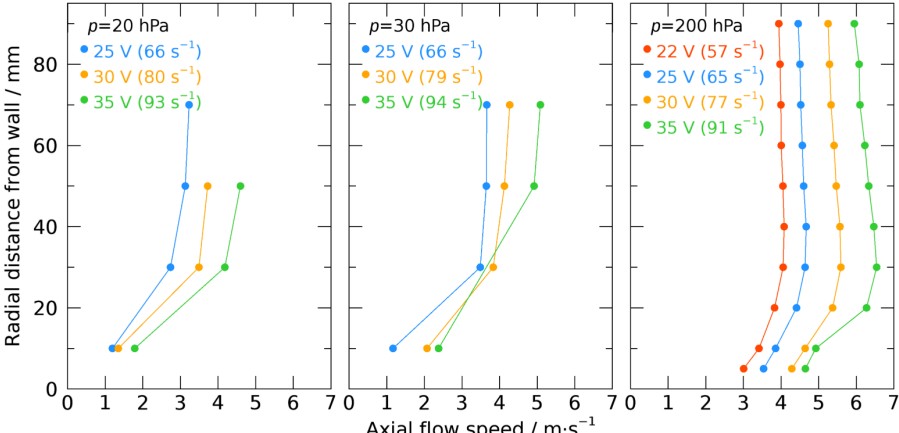

**Figure 3.** Examples of measured radial profiles of axial air flow in the test chamber at a position short before the sensor boom of the radiosonde. The three panels show results for different flow intensities at three different pressure levels. The flow velocities were generated by adjusting the fan voltage and thus its speed (in revolutions per second).

a particle generator (Atomiser aerosol generator, AMT 230) with particle sizes of about 0.2–0.5 $\mu$m, using Di-Ethyl-Hexyl-
Sebacat (DEHS, Topas Co.) as the liquid for the atomiser.

During the measurement of the flow velocities, a RS41 radiosonde is installed inside the quartz tube with the same configuration as for the radiation error experiments, to ensure a consistent flow field around the temperature sensor. At a position along the tube axis which is close to the tip of the sensor boom, the axial and radial components of the air flow are measured at 1 cm to 2 cm intervals in the radial direction from the edge to about the central axis. Thus, this covers half of the tube's diameter, but
cylindrical symmetry is assumed for the flow pattern. The measurements are performed at various pressures and fan rotation speeds.

The LDA measurements show that the magnitude of the radial flow components is, generally speaking, less than approximately 10 % of the axial component, which means that the airflow over the sensor boom is predominantly laminar and in the axial direction, which resembles the situation in flight. Fig. 3 presents examples of the measured flow profiles, which represents
the shape that is expected for a pipe flow. The axial flow velocity is almost zero close to the tube's wall, and increases rapidly to reach a constant value at 3 cm to 4 cm away from the wall. The right-hand panel of Fig. 3 shows a slight decrease of the flow speed near the central axis at higher pressure levels, which is attributed to the radiosonde obstructing the flow.

Below 30 hPa, the injection of the air-particle mixture during seeding causes a noticeable rise in pressure, which presents difficulties in reaching low pressures. Furthermore, at low pressures it becomes increasingly difficult to achieve the minimal
particle density which is required to ensure sufficient LDA counts. As a result, no measurements were performed below 20 hPa.





**Table 1.** Polynomial coefficients used in Eq. (1) for calculation of ventilation speed as function of pressure and fan rotation.

| $i, j$ | $c_{ij}$ | $i, j$ | $c_{ij}$ | $i, j$ | $c_{ij}$ |
|--------|----------|--------|----------|--------|----------|
| 0, 0 | $-1.03154 \times 10^{-6}$ | 0, 1 | $9.07802 \times 10^{-2}$ | 0, 2 | $4.05845 \times 10^{-3}$ |
| 1, 0 | $1.36741 \times 10^{-6}$ | 1, 1 | $-3.85667 \times 10^{-2}$ | 1, 2 | $1.52509 \times 10^{-2}$ |
| 2, 0 | $-2.84620 \times 10^{-7}$ | 2, 1 | $4.87337 \times 10^{-3}$ | 2, 2 | $-1.34985 \times 10^{-3}$ |
| 3, 0 | $1.61256 \times 10^{-8}$ | 3, 1 | $-1.67718 \times 10^{-4}$ | 3, 2 | $4.40378 \times 10^{-5}$ |
| 4, 0 | $-2.70407 \times 10^{-10}$ | 4, 1 | $1.67955 \times 10^{-6}$ | 4, 2 | $-4.77922 \times 10^{-7}$ |

The LDA measurements are used to determine the flow speed around the sensor boom as a function of pressure $p$ and fan rotation speed $f_{\text{fan}}$. The resulting parameterisation of the flow speed $v$ is

$$v(p, f_{\text{fan}}) = \sum_{i,j} c_{ij} \cdot p^{i/2} \cdot f_{\text{fan}}^{j/2}, \tag{1}$$

where the polynomial coefficients $c_{ij}$ follow from a non-linear least squares fit to the measurements at the centre of the chamber
with radial distances between 70 mm and 100 mm. These radii correspond to the area that is covered by the sensor boom of the rotating radiosonde. Table 1 lists the values of $c_{ij}$, and the left panel of Fig. 4 shows the resulting 2-D fit to the data. The blue dots at the bottom represent the data points for $v(p, f_{\text{fan}} = 0) = 0$ that were added to constrain the fit, with the purpose to use Eq. (1) to determine fan speed settings for flow speeds that were not covered by the LDA measurements. The plots in Fig. 4 show a close-to linear dependence of the air speed $v$ on fan rotation, and the left panel also shows a strong decrease in air speed
for pressures below 20 hPa, which is attributed to the reduced efficiency of the fan in this pressure range.

A detailed uncertainty budget for the calculated ventilation speed can not be derived. Instead, a combined overall uncertainty $u(v)$ is set to a fixed value of $0.5\,\text{m s}^{-1}$ ($k = 1$). This value is assumed to take potential components from the LDA-technique and those connected to the least-squares fitting (right panel in Fig. 4) into account. In particular it should include potential uncertainties connected to the positioning and effective flow resistance of the radiosonde's body inside the test chamber. The
latter is expected to be of systematic nature and to dominate the overall uncertainty. A thorough estimate of that uncertainty component can at best be made with very elaborate LDA measurements, for example by varying the position, and size and shape (flow cross section) of the radiosonde, which could not be carried out within this study.

## 2.3 Irradiation

### 2.3.1 Light source

The light source is a 2500 W xenon arc lamp (Osram XBO®2500W/OFR), which has an output spectrum that is, for this application, sufficiently similar to that of the Sun. The spatial inhomogeneity of the output beam at the position of the radiosonde was verified to be less than 1.5 %, using a CMP21 pyranometer (Kipp & Zonen). The temporal stability of the lamp's output is monitored regularly in parallel to the RS41 measurements. Using the same CMP21, the irradiance at two fixed distances

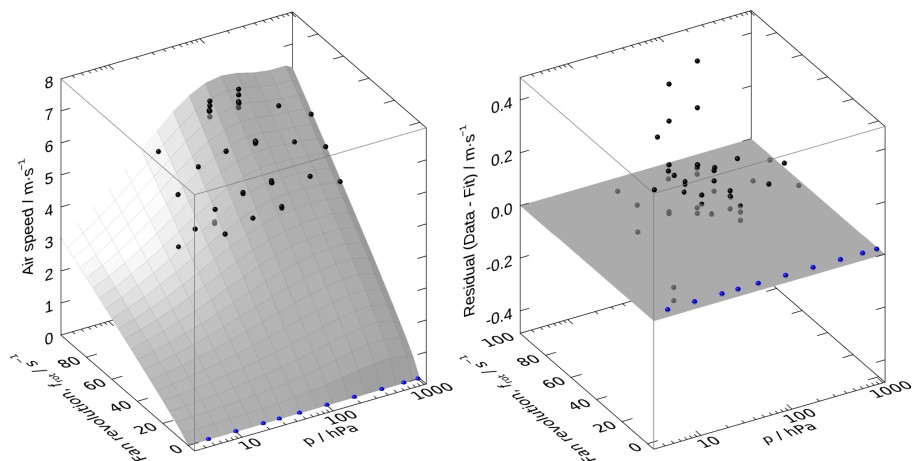

**Figure 4.** Flow speed as function of pressure and fan rotation (in revolutions per second). Left: Symbols denote measured flow speed around the central axis of the test chamber close to the sensor boom of the radiosonde. Blue symbols are manually added points for vanishing flow (fan switched off) to enable reasonable fitting. The surface shows the fit according to the empirical model (Eq. 1). Right: Equivalent plot for the residual, indicating irregular deviations of the data from the fit within about $\pm 0.5\,\mathrm{m\,s^{-1}}$. Note the logarithmic pressure axis for better visibility at low pressures.

($r = 1.0\,\mathrm{m}$ and $r = 1.1\,\mathrm{m}$) is measured in the centre of the beam. These data show a variability (relative standard deviation, $N = 29$) of 1.9 %, with no indications of a significant drift.

### 2.3.2 Flux

Measurements of the flux at various distances from the light source show that the lamp's output decreases with distance following the inverse square-law. The data in Fig. 5 are fitted by

$$I(r) = P_0 \cdot (r - r_0)^{-2}, \tag{2}$$

with $r$ the distance measured from the lamp's housing, and $P_0 = 1444.6\,\mathrm{W}$ and $r_0 = 0.20\,\mathrm{m}$ fit parameters. Here $r_0$ accounts for the position of the virtual image of the $\mathrm{Xe}$ lamp that is generated by the collimating optics. Eq. (2), in combination with the transmission through the quartz wall of the measurement chamber (see Sect. 2.3.3), is used to calculate the flux on the temperature sensor during the experiments.

### 2.3.3 Transmissivity of the quartz tube

The attenuation by the wall of the quartz tube is determined by measuring the flux at the position of the sensor boom with and without the tube in place for various angles of incidence between $0°$ and $60°$. The Fresnel equation, that describes the reflection



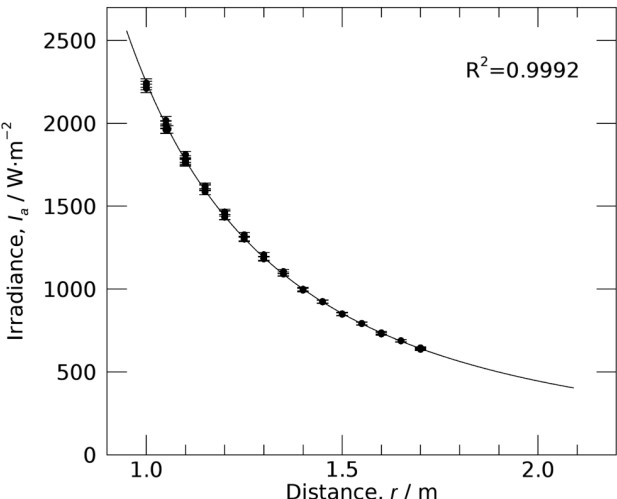

**Figure 5.** Measured irradiance as function of the working distance from the lamp housing. The straight curve denotes a fit according to the quadratic decrease of the irradiance (Eq. 2).

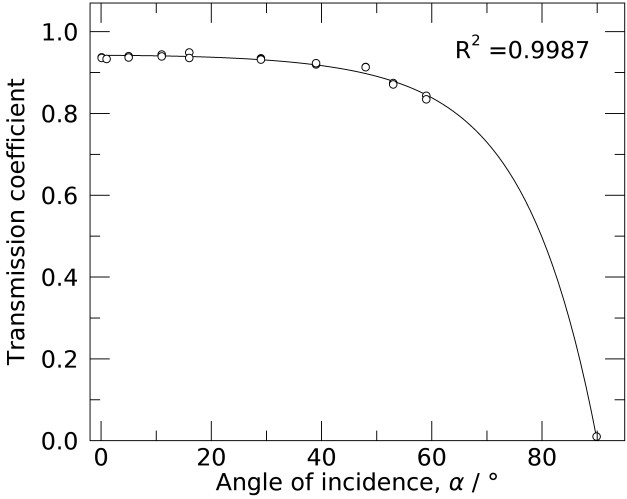

**Figure 6.** Transmission through the quartz glass of the measurement chamber as function of the incidence angle $\alpha$ (see Fig. 8a) as open circles. The curve shows a 2-parameter fit to the data based on Fresnel's laws (Eq. 3).

and transmission of light incident on uncoated surfaces, is used to fit the measured transmittance $T_c$ shown in Fig. 6.

$$T_c(\alpha) = c_0 \cdot \frac{\tan \alpha}{\tan \beta} \cdot \left[ \frac{2 \cos \alpha}{c_1 \cos \alpha + \cos \beta} \right]^2 . \tag{3}$$

Here, $\alpha$ denotes the angle of incidence, and $\beta = \arcsin \frac{\sin \alpha}{n_2}$ follows from Snell's law of refraction, using $n_2 = 1.46$ for the
refraction index of quartz. To aid and constrain the fit, the point $T_c(\alpha = 90°) = 0$ is added, yielding the parameters $c_0 = 0.505$





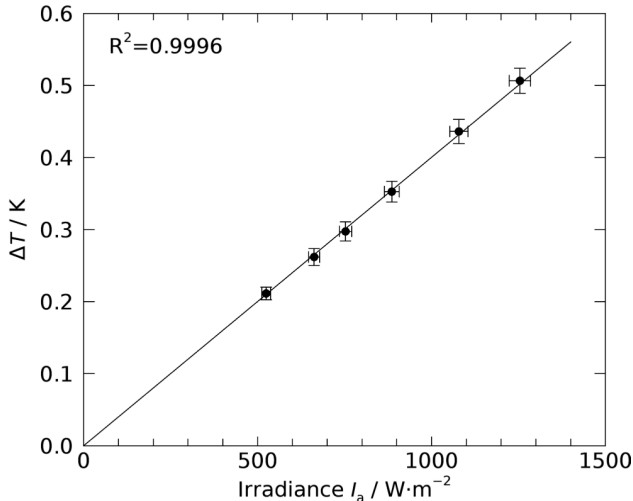

**Figure 7.** Dependence of sensor warming on radiative flux, measured at a pressure of $30\,\mathrm{hPa}$, ventilation speed of $5\,\mathrm{m\,s^{-1}}$, incidence angle $45°$, and rotation period $16\,\mathrm{s}$.

and $c_1 = 0.769$. The attenuation by the window is mainly caused by the reflection from the window's outer surface, and to a lesser extend by absorption or refraction by the quartz glass. As indicated in Fig. 1, a black-coated plate between the radiosonde and the rear wall of the chamber serves as a beam dump and prevents reflections from the rear wall, which would otherwise lead to an overestimation of the radiation error.

### 2.3.4 Linearity of $\Delta T$ with irradiance

The current understanding of the radiative heating of the temperature sensor predicts a linear relationship between flux and the temperature error, when all other parameters are unchanged. This linearity was already proposed in the theoretical approach by Luers (1990) and was confirmed in various experiments such as Lee et al. (2018a, b). This is in agreement with our findings, presented in Fig. 7. Based on the observation that $\Delta T$, and presumably its associated uncertainties, scales linearly with the flux, the measurements are performed for a fixed irradiance level. In practice, the experiments are performed at flux levels between $1025\,\mathrm{W\,m^{-2}}$ and $1142\,\mathrm{W\,m^{-2}}$, which is comparable to the actinic flux at the altitudes where the radiation effect is significant.

### 2.3.5 Extreme solar zenith angle

As mentioned in Sect. 2.1, the angle of incidence $\alpha$ is varied to simulate the conditions for various solar elevation angles. Due to space restrictions, the incidence angle is limited to the range $0°$ to $60°$. In addition, the configuration corresponding to the zenith position of the Sun (incident angle or solar elevation $\alpha = 90°$) can be realised by putting the radiosonde in the fixed position shown in Fig. 8 (b) or (c). Here we exploit the cylindrical symmetry of the situation where the axis of rotation of the spinning radiosonde points towards the Sun. Due to this symmetry, a static radiosonde has the same illumination geometry as





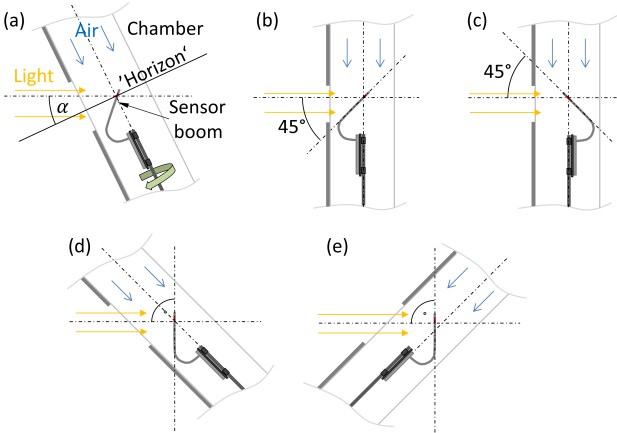

**Figure 8.** Settings for radiation incidence. (a) Configuration for measurements of the radiation error for incidence angles between $0°$ and $60°$, $\alpha$ is equivalent to the sun elevation angle, sonde is continuously rotating around longitudinal axis; (b) and (c) Simulation of incidence with the Sun at its zenith; (d) and (e) Simulation of incidence of diffuse radiation. In (b) to (e) the sonde is fixed (not rotating).

one that is spinning around its axis. Although both sides of the senors boom are assumed to be equally sensitive to radiation, the measurements were performed while illuminating the front (Fig. 8 (b)) and the rear side (Fig. 8 (c)) of the sensor boom, taking the average of both afterwards.

### 2.3.6 Diffuse radiation

Diffuse solar radiation, resulting from light scattered by the air, clouds, or the Earth's surface, can contribute significantly to the total actinic flux. A special configuration of SISTER is used to quantify this contribution to the radiation error, shown in Figs. 8 (d) and (e). Under the assumptions that:

– the effective heating of the temperature sensor depends on the magnitude of the impinging flux only and therefore is the same for direct or diffuse light of the same intensity,

     – the scattered light has a directional uniform distribution,

     – both sides of the sensor boom are equally sensitive to radiation,

the heating of the sensor due to diffuse radiation does not depend on its actual orientation. Therefore, the static, non-rotating,
sensor boom is illuminated perpendicularly with a flux of $527\,\mathrm{W\,m^{-2}}$, which is a good approximation of the diffuse flux encountered in flight. The measurement is performed while illuminating the front (Fig. 8 (d)) and the rear (Fig. 8 (e)) of the sensor boom, and the results are averaged.





**Table 2.** Uncertainty budget for irradiance ($k = 1$). Values are given as relative to applied irradiance.

| Component | Amount |
|---|---|
| Temporal stability Xe-lamp | 1.9 % |
| Light spot inhomogeneity | 1.5 % |
| Distance lamp - radiosonde ($u(r) = 5$ mm) | 1.0 % |
| Transmission through wall of test chamber | 2.0 % |
| Calibration pyranometer | 1.0 % |
| Combined | 3.4 % |

### 2.3.7 Uncertainty budget for irradiance

The contributions to the overall uncertainty of the irradiance at the location of the radiosonde inside the test chamber were
estimated as relative values and are listed in Table 2.

## 3 Experiments

### 3.1 Sequence of measurements

The RS41 is illuminated for typically 30 seconds to three minutes, depending on the actual pressure and ventilation speed, followed by an equivalent cooling phase after closing the shutter. This exposure time is generally long enough for the sensor
temperature to approach the new thermal equilibrium before the shutter is closed again. The rotation of the radiosonde introduces oscillations in the recorded temperature profile, which is caused by the periodically changing exposed surface of the illuminated sensor boom. The analysis of the measurements in Sect. 4.1 shows that the length of the rotation period does not affect the determination of the radiation error. Therefore, a rotation period of 16 s is used for all measurements to represent the spinning of the radiosonde during a typical radiosounding.
The air temperature inside the test chamber is recorded by four thermistors, located at the entrance of the chamber upstream of the radiosonde, that are evenly distributed over the cross section (Fig. 1). These measurements are used to provide the background reference temperature when determining the radiation temperature error.

The measurements are performed for various incidence angles, that correspond to the range of solar elevation angles that occur in real soundings (see Table 3). As a result of the thermal coupling between the sensor and boom, the radiation temperature
error depends on the effectively exposed area of the entire boom, i.e. the exposed surface averaged over a full rotation of the radiosonde. The plot in Fig. 9 shows that this area for a sensor boom tilt of 45°is almost constant for incidence angles below 40°, and increases by almost 50 % between 45°and 90°.

The measurement plan was executed according to the following scheme for a RS41 unit with serial number N4140416:





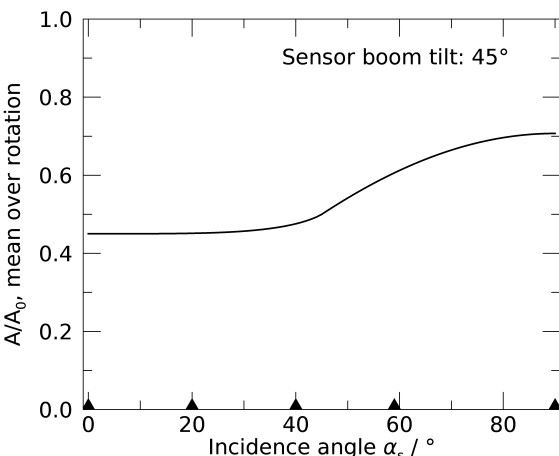

**Figure 9.** Relative exposed sensor boom surface area, averaged over rotation around the vertical, as function of light incidence (solar elevation angle, $\alpha_s$), and for the $45°$ boom angle of the RS41 radiosonde. The triangles mark the angles of incidence in the experiments.

**Table 3.** Adjustable ranges and associated overall uncertainties for the most important experimental parameters

| Parameter | Range | Settings used | Uncertainty |
|---|---|---|---|
| Pressure / hPa | Surf. to 3 | 1000, 400, 200, 100, 50, 30, 20, 15, 10, 7, 5 | 0.4 to 0.6 |
| Ventilation / m s$^{-1}$ | 0 to 7 | max., 6, 5, 4, 3, 2, 1.5, 1 | 0.5 |
| Irradiance / W m$^{-2}$ | 500 to 2000 | 527, 1025 to 1142 | 3.4 % |
| Incidence angle / ° | 0 to 90$^{(*)}$ | 0, 20, 40, 59, 90 | 2 |
| Sonde rotation / s | 1 to $\infty$ | fixed at 16 | – |

$^{(*)}$ (0 to 60)° freely adjustable; 90° fixed

- Set the irradiance by adjusting the distance to the light source (Sect. 2.3.2).

- Set pressure.

- Set the fan rotation speed to generate the desired ventilation speed (between $1\,\mathrm{m\,s^{-1}}$ and $7\,\mathrm{m\,s^{-1}}$).

- Illuminate the radiosonde for ~0.5 min to ~3 min (depending on pressure) with rotating sonde, followed by an equally long measurement with shutter closed; exposure time being an integer multiple of the rotation period.

- Repeat for the pre-defined ventilation speeds (listed in Table 3).

- Repeat for the pre-defined pressure levels (listed in Table 3).

- Repeat for the pre-defined incidence angles (listed in Table 3).





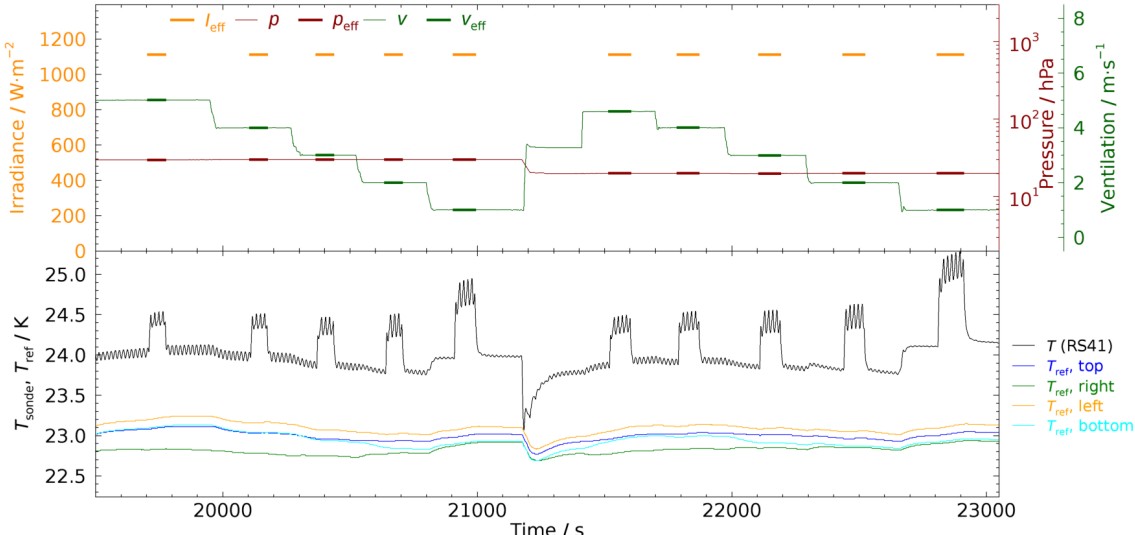

**Figure 10.** Section of raw data for a setup with an incidence of $\alpha = 40°$. Upper panel: Continuous data for pressure and ventilation speed (solid lines); bars indicate values averaged over periods of exposure of the sensor boom for irradiance, pressure, and ventilation. Lower panel: Temperature records of the RS41 radiosonde (black curve), and of the four reference thermistors (coloured curves, with a $-1\,\mathrm{K}$ offset).

The parameter ranges covered with SISTER, and their associated uncertainties, are listed in Table 3. Due to the lack of a pressure gauge of suitable accuracy inside the setup, the reading of the RS41-SGP's internal pressure sensor is used to determine the pressure in the chamber. The uncertainty budget of the pressure reading is based on the manufacturer's specification and on

comparisons that are performed with other pressure sensors. The ventilation speed is set by adjusting the voltage of the fan's power supply, using the voltage-pressure relation of Eq. (1).

The measurement program was confined to one irradiance level for each pressure-ventilation combination because of the linear relation between temperature error and irradiance (Sect. 2.3.4). It takes one day of measurement to cover the pressure-ventilation combinations from Table 3 for each of the eight illumination configurations that are illustrated in Fig. 8.

**4 Evaluation of experimental data**

Figure 10 presents a selected $\sim 1\,\mathrm{h}$ long section of data from a one-day measurement run. Synchronised data from the various sensors are recorded continuously at 1 s-intervals. The exposure periods can be recognised as peaks in the RS41 temperature raw data in the lower panel of Fig. 10. Their magnitudes are from 0.5 K to 1 K in this example. The effect of the continuous rotation of the radiosonde is visible as rapid oscillations superimposed the RS41 temperature curve.





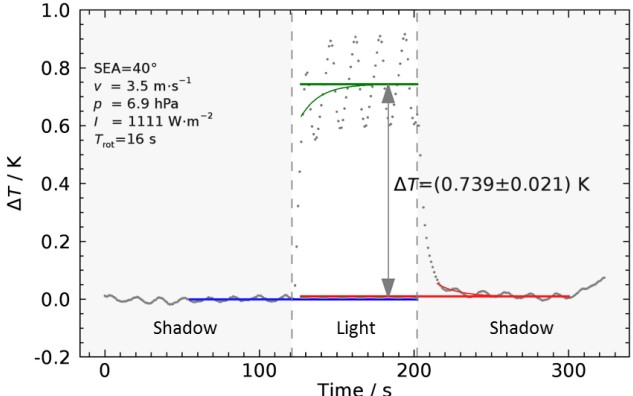

**Figure 11.** Example for the response of the RS41 temperature sensor to irradiation (dotted grey line) after subtraction of the background temperature. Blue line: $T$-mean in the shadow period approximately 1 min prior to light exposure. Green curve: Fit to the $T$-response according to Eq. (4). Straight green line: expected equilibrium value of the $T$-response ($= c_0$). Red curve: Similar fit to the $T$-response after closing the shutter, with the straight red line indicating the re-established shadow temperature level. The oscillating pattern result from the continuous rotation of the sonde with a period of 16 s.

## 4.1 Determination of the solar temperature effect

Temporal fluctuations around the laboratory temperature may occur at the position of the radiosonde, since SISTER is not temperature-stabilised. Such variations are caused by adiabatic effects when setting new pressure levels, such as the temporary drop in temperature at 21200 s in Fig. 10, and at a slower rate due to the heat exchange with the laboratory environment. The fan and the stepper motors, the radiosonde itself, and the energy from the applied radiation are potential local heat sources. These can produce small-scale, temporally-consistent temperature inhomogeneities in the air flow, that are detected by the thermistors that monitor the temperature in the test chamber. The lower panel of Fig. 10 shows that the differences between the individual thermistors due to temperature inhomogeneities can be as large as 0.5 K.

The radiation induced $\Delta T$ is determined from the temperature change measured by the RS41 during exposure, an example of which is shown Fig. 11 for an $\sim$80 s exposure at incidence angle $\alpha = 40°$. After opening the shutter, the measured temperature rises quickly at first, followed by a slower convergence to a quasi-stationary value. In the subsequent analysis, the temperature from one of the four thermistors is subtracted from the radiosonde's readings to compensate for (slow) variations of the background. The choice of this reference thermistor is done manually, selecting the one showing the least of the above-described fluctuations. Before correcting the radiosonde temperature, an offset is added to the thermistor readings, such that the average temperature in the 60 seconds before opening the shutter, represented by the blue line in Fig. 11, matches the average temperature of the radiosonde in that interval. This procedure accounts for possible calibration-related offsets between thermistor and radiosonde sensor. Due to the radiosonde's rotation, the exposed surface of the sensor and the sensor boom changes regularly, causing the oscillating pattern on top of the measured signal. The maxima correspond to the position of the sensor boom with the largest effective area (smallest angle between light beam and the normal to the sensor boom), whereas





the minima correspond to the position where the edge of the sensor boom is turned towards the lamp. In the latter case the

cylindrical temperature sensor is still illuminated, so that $\Delta T$ does not approach zero. These oscillations are a clear proof of

the thermal coupling between the sensor and the sensor boom. Due to thermal inertia of the sensor boom, there may be a time

lag between the oscillation peaks and the actual positions corresponding to minimum and maximum illumination. The lower

panel of Fig. 10 shows that small oscillations are also observed in the shadow phases. These are attributed to the rotating sensor

boom moving through small, persistent temperature inhomogeneities inside the measurement chamber.

During the slow convergence, the temperature sensor approaches thermal equilibrium following an exponential decay, with

the rotation-induced oscillations superimposed. $\Delta T$ follows from fitting this exponential decay with Eq. (4) below, where

the length of the fit interval is chosen to contain an integer number of oscillation periods, to minimise their influence on the

resulting fit, which is represented by the green trace in Fig. 11.

$$\Delta T = c_0 + c_1 \exp(-c_2 t) \tag{4}$$

Here $t$ is the time since opening the shutter, $c_0$ and $c_2$ are auxiliary fit parameters, and $c_0$ represents the new equilibrium

temperaturem which is represented by the horizontal green line in Fig. 11. The fit windows are set manually for each exposure,

because the large variety in shape, size, and temporal behaviour of the response curves for the various experimental settings

would unnecessarily complicate an automated algorithm.

After closing the shutter, the indicated temperature returns to the background level, again following an exponential decay.

Eq. (4) is used again to determine the radiosonde temperature without illumination. The actual resulting $\Delta T$ is determined as

the difference between the fitted equilibrium temperature rise under illumination (the green line in Fig. 11), and the average of

the two background temperatures before and after the exposure period, that are indicated by the blue and red lines in Fig. 11,

respectively.

Another advantage of the fitting procedure is that it is possible to determine $\Delta T$ even when the thermal equilibrium is not

reached during the exposure. This allows to keep the exposure periods reasonably short, especially in case of low pressure and

ventilation speed, where it takes several minutes to reach the equilibrium.

The resulting $\Delta T$ does not depend on the accuracy of the absolute temperature measurement of the radiosonde and the

reference thermistors, because the evaluation relies on differences. As a result, the uncertainty from that analysis step is small

(see Fig. 11 and Sect. 4.3), because it depends mainly on the reproducibility of the indicated sonde temperature before and after

exposure, and to a small extent on the formal uncertainty of the fit parameter $c_0$ or the sensor intrinsic noise and resolution.

The larger final uncertainty is associated with the systematic uncertainties of the experimental pressure and ventilation values

to which the actual $\Delta T$ is assigned (Sect. 4.3).

## 4.2  Data reduction and evaluation

Both pressure (or air density) and air speed determine the efficiency of the ventilation and thereby the exchange of sensible heat

between the surface of the sensor and sensor boom and the ambient air. This heat exchange depends on the thermodynamic and

fluid-dynamic properties of air, and also in a complex way on the turbulence conditions around the sensor, which in turn are





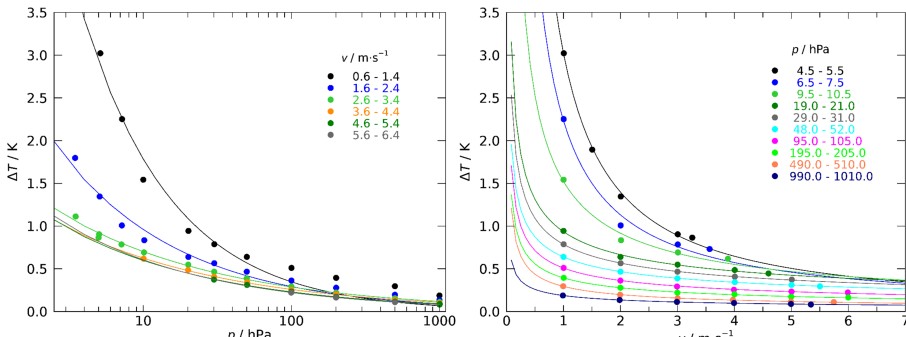

**Figure 12.** Example of measurement results (symbols), for a setting with an irradiation of $1024\,\mathrm{W\,m^{-2}}$, an incidence angle of $59°$, and continuous sonde rotation with a period of $T_R = 16\,\mathrm{s}$. Left panel: $\Delta T$ vs. $p$; $v$ as parameter. Right panel: $\Delta T$ vs. $v$; $p$ as parameter. Pressure is given on a logarithmic scale for better visibility. The solid curves represent fits to the data using $y = a \cdot x^{-b}$, the values of $b$ are between 0.30 and 0.65.

influenced by the sensor shape and the flow direction (see e.g. Luers, 1990). In our analysis we use a simplified parameterisation of the dependence of $\Delta T$ on the ventilation parameters $p$ and $v$, that relies on the well known effect that the convective heat transfer is more efficient for higher values of $p$ and $v$.

It is found that for most cases the dependence of $\Delta T$ on $p$ or $v$ follows a $x^{-b}$-relation, where the value of $b$ lies between $0.30$ and $0.65$. The curves in Fig. 12 indicate this for pressures $> 20\,\mathrm{hPa}$ and typical in-sounding ventilation speeds of $> 3\,\mathrm{m\ s^{-1}}$. For lower pressures or ventilation speeds, this relation no longer holds as is illustrated by the significantly faster rise of $\Delta T$ (see blue and black lines in the left panel).

A two-dimensional polynomial function for $\Delta T$, dependent on the square root inverse of the pressure $(1/p)$ and ventilation
speed $(1/v)$, is fitted for each incidence angle to the measured $\Delta T(p,v)$-data set:

$$\Delta T(p,v) = \sum_{i,j} c_{ij} \left(\frac{1}{\sqrt{p}}\right)^i \left(\frac{1}{\sqrt{v}}\right)^j. \tag{5}$$

Prior to fitting, data points with $\Delta T = 0.01\,\mathrm{K}$ at $v = 50\,\mathrm{m\ s^{-1}}$ are added over the entire pressure range (not shown in Fig. 13), for all data sets. This ensures that the fit is constrained so that the results comply with the expectation of a smooth and monotonous decrease of $\Delta T$ at large $v$, and allows estimating reasonable values in the high-ventilation and low-pressure range
where no measurement data are available.

In the left panel of Fig. 13, the fit according to Eq. (5) is visualised as grey surface, based on the same data as in Fig. 12. The red-shaded part of the surface indicates the range of realistic ventilation speeds occurring during real soundings. The blue surfaces in the right panel represent uncertainty estimates, and are discussed in Sect. 4.3. The visualisation in Fig. 13 is intended to give an overview of the distribution and scatter of the data points.

The fits were created for all of the eight illumination configurations, i.e. for $0°$, $20°$, $40°$, $59°$, and the two realisations of both the zenith setup $(90°)$ and the simulation of diffuse radiation, respectively. The two $90°$ and 'diffuse' results were





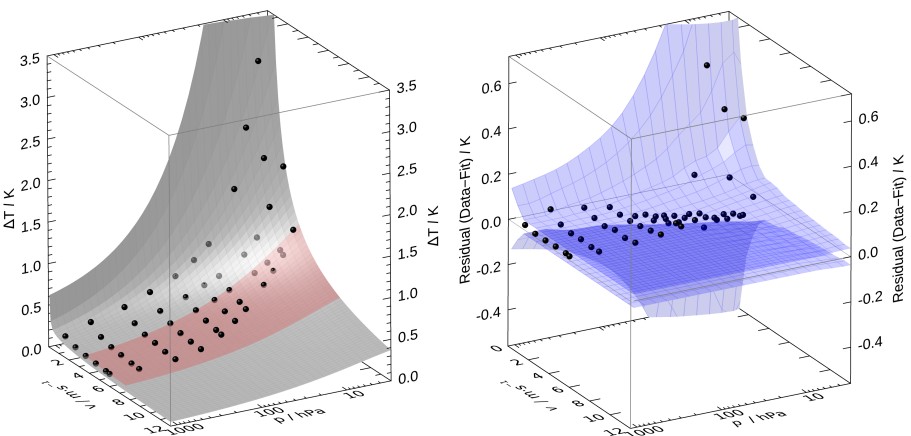

**Figure 13.** Left: Three-dimensional representation of the measured data (black dots) and the fitted model according to Eq. (5) for the same data set as in Fig. 12. The red coloured part of the surface denotes the typical $v$-range in radiosonde ascents. Right: Overall uncertainty in a similar representation, showing the deviations of the measured points from the model as black dots, and $1\sigma$-uncertainty estimates as blue shaded surfaces.

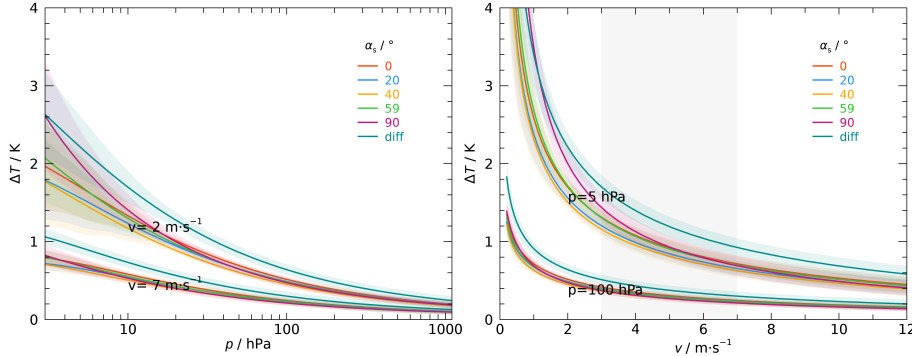

**Figure 14.** Comparison of the solar temperature error for different radiation incidence angles ($\alpha_s$). $\Delta T$ is normalised to an irradiance equivalent to the solar constant ($1361\,\mathrm{W\,m^{-2}}$) beforehand. Left: Examples for ventilation levels of $2\,\mathrm{m\,s^{-1}}$ and $7\,\mathrm{m\,s^{-1}}$. Right: Examples for pressure levels of $5\,\mathrm{hPa}$ and $100\,\mathrm{hPa}$. Shaded areas around the graphs denote $1\sigma$-uncertainty estimates. The grey shading in the background indicates the range of prevailing ventilation velocities in soundings.

each averaged as explained in Sect. 2.3.5 and 2.3.6. The parameterisation of Eq. (5) is used in the operational solar radiation correction that is described in Sect. 6.1.

The irradiances are slightly different for the data sets of the different incidence angles. Using the proportionality of $\Delta T$
with the amount of radiation, the results of these runs can be compared after normalisation to an irradiance reference value. Figure 14 shows example data at two ventilation and pressure levels in this pre-evaluated form, where the $\Delta T$ from Eq. (5) are normalised to $1361\,\mathrm{W\,m^{-2}}$. The plots disclose that the $\Delta T$ for angles between $0°$ and $59°$ are consistent. That is, the





temperature error does not vary systematically with incidence in that range. The 90°-curves (Sun at its zenith) show somewhat increased values at both low pressure ($<10\,\mathrm{hPa}$) and ventilation speed ($<3\,\mathrm{m\,s^{-1}}$). Thus, the measurement results show a fair

overall compatibility with the simple geometry-based model presented in Fig. 9.

Figure 14 includes curves labelled 'diff', which lie – to a large extent significantly – above the group of curves for the different incidence angles. These are the results from the particular experimental configuration where the effect of diffuse radiation is simulated. The heating of the temperature sensor is maximal when the sensor boom is irradiated perpendicularly without rotation, as expected. Translated to the in-flight conditions that means that the temperature effect by diffuse radiation

corresponds to a perpendicular incidence of direct radiation with the same irradiance (radiant flux received by a surface per unit area), and therefore represents the maximum achievable effect, since the heat flux by diffuse light to the sensor is essentially independent on its orientation (no cosine-effect). This is accordingly taken into account in the derivation of the operational radiation correction in the GRUAN RS41 processing.

## 4.3 Uncertainties

The uncertainty of the measured temperature error, $u(\Delta T)$, is a combination of four main components:

- Uncertainty associated with the analyses of individual radiation-induced temperature leaps, see Fig. 11. This component consists of several sub-components, including the uncertainty of the parameters $c_0$ for both the 'up' and 'down' fits to the temperature leap flanks using Eq. (4), the random $T$-noise during the period before shutter opening, and the offset between the two shadow temperatures 'enclosing' the light phase. Its magnitude is generally small and considered

uncorrelated.

- Uncertainty connected to radiation ($3.4\,\%$, $k = 1$, see Sect. 2.3.7, uncorrelated). Due to the linear relationship between irradiance and $\Delta T$, this component is equivalently given as relative.

- Uncertainty of pressure ($0.4\,\mathrm{hPa}$ – $0.6\,\mathrm{hPa}$), considered systematic (correlated). Pressure uncertainty is converted to temperature uncertainty via the local sensitivity ($\frac{\partial(\Delta T)}{\partial p}$) from Eq. (5).

- Uncertainty of ventilation speed, set to a constant value of $0.5\,\mathrm{m\,s^{-1}}$ ($k = 1$), and considered systematic (correlated). Ventilation uncertainty is converted to temperature uncertainty via the local sensitivity ($\frac{\partial(\Delta T)}{\partial v}$).

Ventilation speed is the strongest contributor to the combined uncertainty.

The results for the two respective realisations (front and back side of the sensor boom) of both the 'zenith' and the 'diffuse' settings are averaged before use in the further processing, as noted in Sects. 2.3.5 and 2.3.6. The uncertainty of these averages

is estimated as the combination of the individual uncertainties, plus another component calculated with $u(\Delta T)_{\mathrm{av}} = |(\Delta T)_a - (\Delta T)_b|/(2\sqrt{3})$. The latter assumes that the 'true' results lie somewhere in between those for the two realisations.

The overall uncertainties $u(\Delta T)$ are stored in $(21 \times 21)$ point data-arrays in terms of pressure and ventilation for each of the eight experimental settings. The values for the pressure dimension of these arrays are defined between $2.72\,\mathrm{hPa}$ and $1096.63\,\mathrm{hPa}$ with logarithmic spacing; the ventilation is defined linear with $0\,\mathrm{m\,s^{-1}}$ to $12\,\mathrm{m\,s^{-1}}$. The arrays are the result of





| Altitude range / km | Resolution / m |
| --- | --- |
| 0–5 | 250 |
| 5–21 | 500 |
| 21–36 | 1000 |
| 36–52 | 2000 |

**Table 4.** Vertical resolution of the Streamer model.

an interpolation of the above discussed uncertainties at the $(p, v)$-positions of the measured $\Delta T$ by using a Kriging procedure. Before interpolating, points are added for $u(\Delta T)$ in the $(p, v)$-plane where no measured $\Delta T$ were available, with a spacing similar to that of the measured data. Their magnitude is fixed to 10 % of the $\Delta T$ calculated with Eq. (5). At $v = 0.1\,\mathrm{m\,s^{-1}}$ and extending over the whole pressure range, further points are added to populate the edge of the $(p, v)$-plane where ventilation approaches zero. The values are defined as three times the $\Delta T$ at $v = 1\,\mathrm{m\,s^{-1}}$, which mimics the expected sharp increase of

$u(\Delta T)$ at vanishing flow speed. Additionally, an absolute lower boundary of 0.03 K is defined for $u(\Delta T)$. This procedure is justified as practical approach to get meaningful results from the Kriging, and thus to provide useful uncertainties also for those $(p, v)$-ranges, which are not covered by the measured data.

     An example for the combined uncertainties of $\Delta T$ is visualised in the right panel of Fig. 13. The black dots show the measured $\Delta T$ as deviation from the model fit, and the blue surfaces denote the $u(\Delta T)$-array as $\pm 1\sigma$ uncertainties. They do not

exceed $\sim$0.2 K at the lowest pressures and for ascent-realistic ventilation speeds higher than about $3\,\mathrm{m\,s^{-1}}$, and the majority of the residual points are within the volume spanned by the uncertainty surfaces. The plot shows that the measurement uncertainty increases significantly at low pressures ($<$10 hPa) and especially at low ventilation speed ($<$2 m s$^{-1}$). Another example is shown in Fig. 14, with the shaded areas indicating $1\sigma$ uncertainty bands. Similarly to the $\Delta T$, the uncertainties were linearly scaled for the actual radiation correction to match the different atmospheric irradiance levels.

**5   Simulation of the radiation field**

The actinic flux is needed to relate the laboratory characterisation of the temperature sensor's susceptibility to radiation to the actual temperature error. Since coincident in situ measurements of the solar radiation at the time of the radiosounding are not available, the actinic flux is estimated using the radiative transfer model (RTM) Streamer (see Key and Schweiger, 1998; Key, 2002). The Streamer model was designed to calculate broadband up- and down-welling fluxes in 24 shortwave and 105

longwave bands for a wide range of atmospheric conditions, using a plane-parallel atmosphere and a flat surface. The model supports surface albedo, gas absorption, and scattering from ice, water and mixed-phase clouds, for atmospheric profiles up to 100 layers. The model's fast code is integrated in the GRUAN processing to perform online calculations of radiation profiles. As mentioned before, the GRUAN radiation correction only considers the shortwave fluxes.



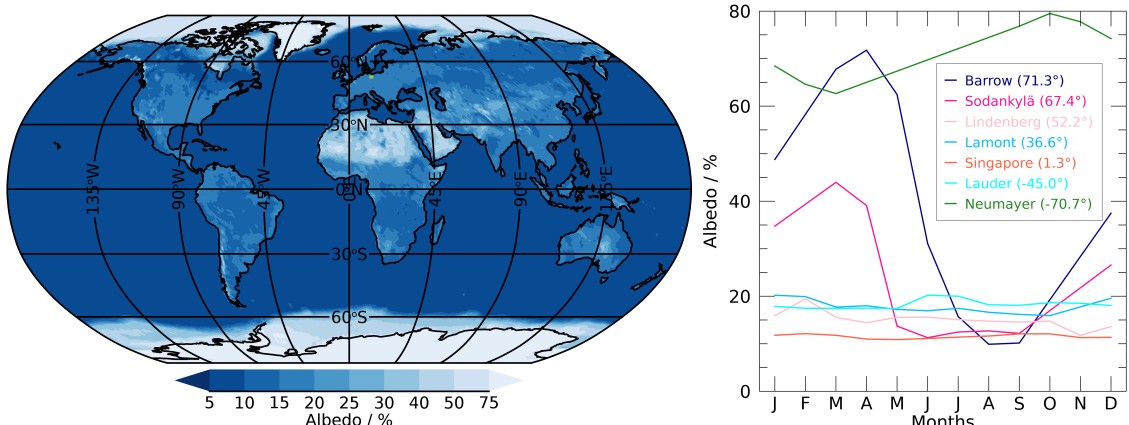

**Figure 15.** Left: Map of the monthly-averaged global surface albedo between $0.25\,\mu m$ and $2.5\,\mu m$ for the month of June, used for the processing of a radiosounding performed at Lindenberg on 11 June 2020, 12:00 UTC. Right: Annual cycle of the monthly averaged albedo at GRUAN sites of various geographical latitudes.

## 5.1 Model input parameters

The model's input profiles of pressure, temperature, and relative humidity are taken from the actual radiosonde measurement. Information on the surface albedo is taken from a global data set, and a generic, representative cloud scenario is defined by cloud layers close to the surface and the tropopause, respectively.

The radiosonde profile, that has a vertical sampling of $\sim5\,m$, is mapped on the much coarser resolution of the model by selecting the data point with the nearest altitude for each of the 100 model-layers. The vertical resolution (thickness of the

model layers) depends on altitude, and is highest at the surface (see Table 4). The measured radiosonde profiles are extended to 52 km by inserting a standard profile above the burst point (see chapter 15 in Key, 2002). Latitude-dependent standard profiles are available for the tropics ($\phi < 23.43°$), mid-latitudes (summer and winter, $23.43° \leq \phi < 66.56°$), and the Arctic ($\phi \geq 66.56°$).

A map that incorporates the spatial and temporal variability of the surface albedo is derived from a satellite-based data set

provided by Karlsson et al. (2017). Data from January 2005 to December 2015 are used to generate monthly averages of the broadband surface albedo between $0.25\,\mu m$ and $2.5\,\mu m$, at a spatial resolution of $0.25° \times 0.25°$ (shown in Fig. 15). The surface albedo that is used in the RTM calculations is the arithmetic mean of the pixels that contain the radiosonde's trajectory.

### 5.1.1 Cloud scenario

Owing to their high reflectivity and light scattering properties, clouds can considerably alter the effective albedo of a scene. The

inherently large spatio-temporal variability of the cloud cover, without having accurate information on its actual configuration, constitutes a substantial source of uncertainty when estimating the actinic flux on the radiosonde. As a compromise, a radiation profile which is the average of two extremes, a cloudy and a clear-sky situation, is constructed for each radiosounding.





**Table 5.** RTM input parameters for cloud layer scenario.

| Parameter | Layer 1 | Layer 2 |
|---|---|---|
| Phase | liquid | ice solid column |
| Cloud optical thickness | 15.7 | 1.1 |
| Water content | $0.1\,\mathrm{g\,m^{-3}}$ | $0.01\,\mathrm{g\,m^{-3}}$ |
| Top of cloud | $3.0\,\mathrm{km}$ | tropopause |
| Cloud thickness | $1.0\,\mathrm{km}$ | $2.0\,\mathrm{km}$ |

The cloudy scenario consists of two separate cloud layers, whose properties are listed in Table 5: an optically thick, wet, low-level cloud at 3 km above the surface, and an ice-cloud underneath the tropopause. The latter requires tropopause detection by the data processing. In case of an early balloon burst below the tropopause, a tropopause altitude of 7.5 km is assumed. For the clear-sky case the same parameter settings for the model atmosphere are used, but without the cloud definitions.

### 5.1.2 Solar elevation angle during ascent

The radiation field depends strongly on the solar elevation angle $\alpha_s$, and since the solar elevation changes continuously with time, sometimes augmented by the horizontal drift of the balloon, this angle needs to be calculated for every data point of the ascent. For accurate evaluation of $\alpha_s$, the GRUAN data processing uses the Solar Position Algorithm (SPA) by Reda and Andreas (2003, 2004).

Quick changes of the solar radiation occur at sunrise or sunset, when the Sun crosses the horizon. However, at small negative values for $\alpha_s$, i.e. if the Sun is just below the horizon for an observer on the ground, the Sun can still illuminate the radiosonde at higher altitudes. The shift of the day/night limit with altitude is illustrated in Fig. 16, which shows that sunset ($\alpha_s = 0$) occurs for the ground-based observer when the radiosonde is at 5 km altitude, but that the radiosonde was illuminated until 11.5 km altitude, when the Sun was already more than $4°$ below the horizon. Because Streamer assumes a plane parallel surface and atmosphere, negative $\alpha_s$ are associated with the nighttime situation. To resolve this, $\alpha_s$ is transformed to a small positive value, equivalent to the Sun being just above the horizon, as experienced by the radiosonde.

The transformation of $\alpha_s$, respectively solar zenith angle $\theta_s$, goes as follows: Assuming the Earth to be a perfect sphere, the angle below the horizon at which the centre of the solar disc is still visible at altitude $h$ is given by

$$\alpha_{\mathrm{cor}} = \arccos \frac{R}{R+h}, \tag{6}$$

with $R$ the Earth's radius. Using $R = 6.371 \cdot 10^6$ m, this can be approximated by

$$\alpha_{\mathrm{cor}} = 0.0321 \cdot \sqrt{h}, \tag{7}$$

with $h$ in metres. The transformed zenith angle $\theta_s'$ is given by

$$\theta_s' = \theta_{s,\mathrm{lim}} + (\theta_s - \theta_{s,\mathrm{lim}}) \cdot f_{\mathrm{hc}}. \tag{8}$$





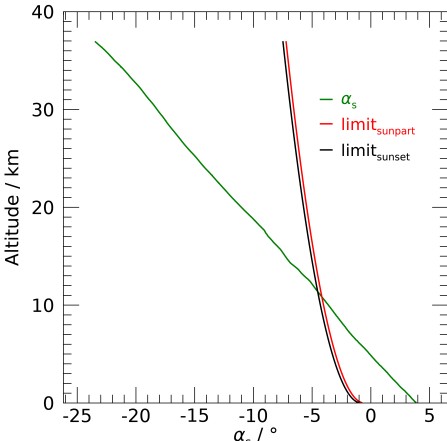

**Figure 16.** Example of solar elevation angle $\alpha_s$, and assignment to daylight phases (day, night), for a sunset sounding in Singapore (2020-05-09, 12:00 UTC). The radiosonde is launched ($h = 0$ km) short before sunset ($\alpha_s = 4°$). At about 11.5 km the Sun is at first partially obscured when passing the horizon ($\text{limit}_{\text{sunpart}}$), and sets completely shortly afterwards ($\text{limit}_{\text{sunset}}$). Darkness is assumed for the remaining part of the ascent, i.e., twilight is considered insignificant for the radiation error.

with $\theta_{s,\text{lim}} = 85°$, and

$$f_{\text{hc}} = \frac{90 - \theta_{s,\text{lim}}}{90 - \theta_{s,\text{lim}} + \alpha_{\text{cor}}}. \tag{9}$$

This 'horizon correction' is applied to zenith angles $\theta_s$ exceeding $\theta_{s,\text{lim}}$, which means that it is also applied when the Sun is still above the horizon. This compensates Streamer's overestimation of the absorption at low $\alpha_s$ due to the plane parallel atmosphere.

## 5.2 Simulation and results

A typical radiosonde ascent takes about 90 minutes to reach the balloon burst point, during which horizontal drifts of more than 100 km from the launch point are not unusual (Seidel et al., 2011). Depending on location and time of the ascent, the solar elevation angle $\alpha_s$ can vary over more than 15° during the flight. To account for the effect of these changes on the radiation field, Streamer calculations are performed for the range of solar elevation angles encountered in flight at steps of 0.5°. For each step, the cloudy and the cloud-free scenario are simulated. To reduce the computational effort, the simulation depth and accuracy is reduced by using the simple two-stream solver. The simulated profiles for each solar elevation angle are stored in a look-up table (LUT). Radiation flux components are estimated for each data point of the ascent by linear interpolation from this LUT.

The Streamer model produces profiles of downward direct ($I_{\text{dir}}$), and diffuse upward ($I_{\text{dif,up}}$) and downward ($I_{\text{dif,down}}$) radiation for both the cloudy and the clear-sky scenarios. The diffuse radiation used for the final temperature correction is the sum

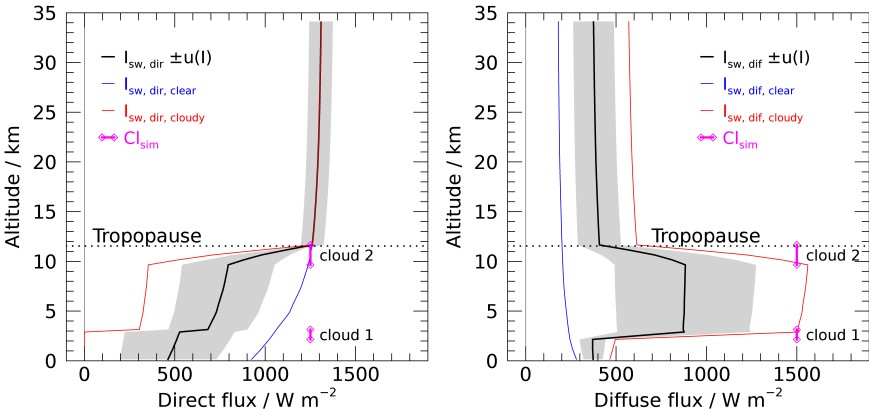

**Figure 17.** Example of simulated solar radiation fluxes including uncertainty estimates (Lindenberg, 2020-06-11, 12:00 UTC, solar elevation angle ∼59°). Left: direct solar radiation. Right: diffuse radiance from surface albedo, and cloud reflection and scattering.

of the upward and downward components:

$$I_{\text{dif}} = I_{\text{dif,down}} + I_{\text{dif,up}}, \tag{10}$$

under the assumption that both sides of the sensor boom are equally sensitive to solar radiation (Sect. 2.3.6).

The radiation flux profiles $I_{\text{dir}}$ and $I_{\text{dif}}$ are constructed from the average of their respective cloudy (cl) and clear-sky (cs) cases:

$$I_{\text{dir}} = \frac{1}{2}(I_{\text{dir,cs}} + I_{\text{dir,cl}}), \qquad I_{\text{dif}} = \frac{1}{2}(I_{\text{dif,cs}} + I_{\text{dif,cl}}), \tag{11}$$

with the associated $k = 1$ uncertainties for a rectangular a priori distribution bound by both extremes:

$$u(I_{\text{dir}}) = \frac{|I_{\text{dir,cs}} - I_{\text{dir,cl}}|}{2\sqrt{3}}, \qquad u(I_{\text{dif}}) = \frac{|I_{\text{dif,cs}} - I_{\text{dif,cl}}|}{2\sqrt{3}}. \tag{12}$$

To account for the reduced model-accuracy in favour of faster computing time, a lower limit of 5 % was imposed on the uncertainty in $I_{\text{dir}}$ and $I_{\text{dif}}$:

$$u_{\min}(I_{\text{dir,dif}}) = 0.05 \cdot I_{\text{dir,dif}}, \tag{13}$$

based on sensitivity tests of the Streamer model. Because of the above-mentioned limitations of Streamer, $u_{\min}$ is doubled for solar elevation angles smaller than 5°.

Fig. 17 shows an example of profiles of simulated direct and diffuse solar radiative fluxes for the cloud and clear-sky scenarios, as well as the mean across both (black lines), and the uncertainty estimates (shaded areas, see Sect. 6.2). The employed cloud model, with a layer at a few kilometres above ground and another one just below the tropopause, leads to considerably different radiation fluxes compared to the clear-sky case. In particular, multiple scattering significantly amplifies the diffuse radiation in the altitude range between the cloud layers. Therefore, especially in that altitude range in the troposphere, the





simulated fluxes from the two scenarios may be seen as extremes for the range of radiative fluxes that may occur for real cloud configurations. The uncertainty of the simulated radiation fluxes is essentially derived from the difference of these boundary scenarios, and therefore significantly larger in the troposphere than in the stratosphere.

## 6     RS41 solar radiation correction and implementation in the GRUAN data product

The correction algorithm calculates the bias $\Delta T_{\mathrm{rad}}$ for each data point by combining the experimentally derived radiation
sensitivity of the temperature sensor $\Delta T(p, v)$ from Eq. (5) and the modelled radiation profiles given by Eq. (11). The total sensor heating is then evaluated as the sum effect of the components from the diffuse and the direct radiation.

### 6.1     Correction

#### 6.1.1     Direct radiation

The direct radiation $I_{\mathrm{dir}}$ is provided by the RTM on the altitude grid of the sounding profile. The input parameters $p$, ventilation
speed $v$, and the solar elevation angle $\alpha_s$, are derived from the actual sounding data, where $\alpha_s$ follows from the time of the measurement, GPS altitude, and geographical position of the radiosonde. The pressure $p$ is calculated based on GPS altitude and the temperature and humidity profiles under the assumption of hydrostatic equilibrium. The ventilation speed is calculated by combining the ascent speed, which is derived from GPS altitude changes, and an effective horizontal speed component, which is caused by the pendulum-related oscillations around the mean horizontal trajectory. The temperature effect $\Delta T_{\mathrm{dir,exp}}$
(Eq. 5) due to direct radiation is experimentally determined for five different solar elevation angles $\alpha_{s,\mathrm{exp}}$. These are used as support points for the determination of the final temperature bias $\Delta T$ at any given angle by nearest-neighbour interpolation. Thus, the part of the bias associated with direct radiation is

$$\Delta T_{\mathrm{dir}}(p, v, \alpha_s) = \frac{I_{\mathrm{dir}}}{I_{\mathrm{exp}}} \cdot \Delta T_{\mathrm{dir,exp}}(p, v, \alpha_s), \tag{14}$$

using the linear response for scaling the temperature effect from the fixed experimental radiation settings $I_{\mathrm{exp}}$ to the modelled
radiation $I_{\mathrm{dir}}$.

#### 6.1.2     Diffuse radiation

Similar to the direct radiation, $I_{\mathrm{dif}}$ is provided by the Streamer model on the altitude grid of the sounding profile, and $p$ and $v$ are taken from the actual sounding data as described in Section 6.1.1. The solar elevation angle $\alpha_s$ is not needed in the calculation of $\Delta T_{\mathrm{dif}}$ because of the aforementioned anisotropy of the diffuse radiation. Instead $\Delta T_{\mathrm{dif}}(p, v)$ follows from Eq. (5), using the
coefficients derived for diffuse experimental setup and by linear scaling with the modelled diffuse flux:

$$\Delta T_{\mathrm{dif}}(p, v) = \frac{I_{\mathrm{dif}}}{I_{\mathrm{exp}}} \cdot \Delta T_{\mathrm{dif,exp}}(p, v). \tag{15}$$



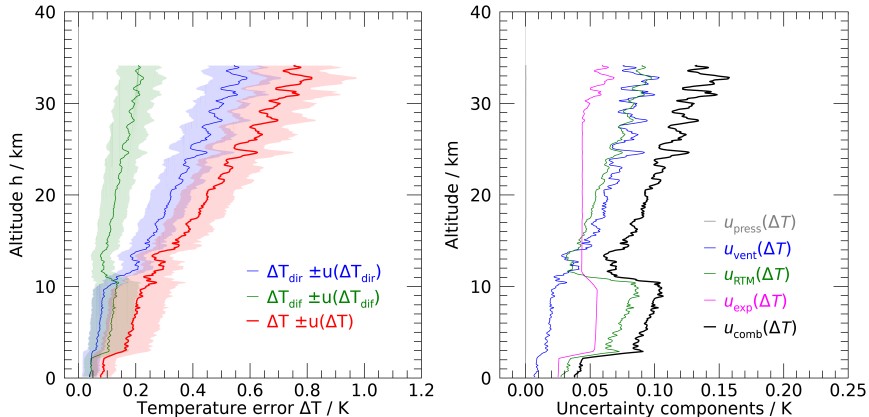

**Figure 18.** Left: Example for the estimated solar radiation error of temperature measurements (Lindenberg, 2020-06-11, 12:00 UTC). Red: Overall bias, blue: component from direct radiation, green: component from diffuse radiation; shaded areas indicate uncertainties ($k = 1$). Right: Uncertainty components ($k = 1$) associated with pressure, ventilation speed, radiation model, and laboratory experiments.

### 6.1.3 Overall radiation correction

The calculated radiation error, which is subtracted from the raw temperature profile, for each data point is given by

$$\Delta T_{\text{rad}} = \Delta T_{\text{dir}} + \Delta T_{\text{dif}}, \tag{16}$$

yielding the corrected temperature

$$T_{\text{rc}} = T_{\text{raw}} - \Delta T_{\text{rad}}. \tag{17}$$

The left panel in Fig. 18 shows a typical profile of the estimated radiation error and the contributions from diffuse and direct radiation for a noon sounding at a mid-latitude site in summer. The total bias (red curve) is approximately 0.1 K at ground level

and increases to more than 0.2 K towards the tropopause (at approximately 11 km), making it a non-negligible effect throughout the troposphere. The discontinuities at (2 to 3) km and 11 km are caused by the two cloud layers used in the radiation model. As a result of the enhanced diffuse radiation between these cloud layers, the diffuse portion of the radiation bias (green trace) exceeds that for direct radiation (blue trace) in the troposphere. Above the tropopause, the bias increases steadily to reach approximately 0.8 K at 35 km, with the portion from direct radiation being the dominant contribution.

Above ~25 km, large-scale oscillations occur, which can be observed in numerous soundings. These are attributed to ascent speed variations caused by gravity waves. The calculated ventilation speed $v$ that is used in the radiation correction is based on the ascent speed, which in turn is derived from the GPS measurements. In case of gravity waves or other vertical air movements, this ascent speed can be different from the ventilating air speed at which the balloon rises through the air.

Uncertainties, that are further discussed in Sect. 6.2, are indicated in Fig. 18 as shaded areas in the left panel, and as graphs

for individual components in the right panel. The uncertainty due to pressure is less than 0.01 K over the entire profile, whereas



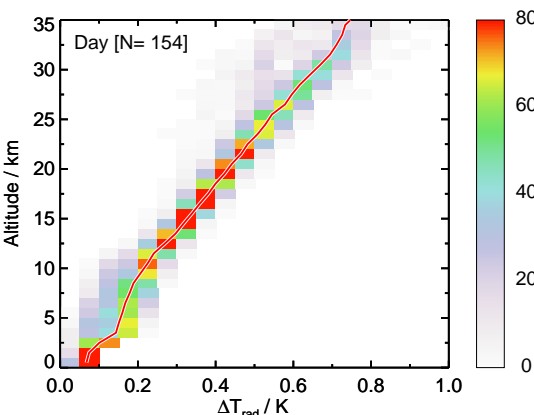

**Figure 19.** Scatter density plot of the estimated radiation temperature error $\Delta T_{\rm rad}$ for 154 daytime radiosounding profiles performed in Lindenberg between 2014 and 2021. The colour represents the number of data points in $0.05\,\mathrm{K}\times1\,\mathrm{km}$ wide bins, and the red line denotes the median of the bias in each bin.

above the tropopause the contributions from ventilation, radiation modelling, and laboratory experiments are of of comparable magnitude. The uncertainty in the troposphere is dominated by the uncertainty in the estimated radiation profile, which is caused by the absence of information on the actual cloud configuration at the time of the sounding.

Fig. 19 shows that the ensemble of estimated error profiles for a set of 154 daytime soundings is, on a first approximation,
fairly similar to the individual profile shown in Fig. 18. The median (red trace in Fig. 19) exhibits the same trend in the stratosphere, with nearly the same value at 35 km (approximately 0.75 K), and the oscillations that are attributed to gravity waves are obviously absent due to the averaging. The slight spread in the data above 25 km is possibly due to seasonal variations of the solar elevation angle, but this is subject of further investigation. The increased variability of the data between 2 and 12 km is also attributed to the seasonal spread in the radiation profile due to the strong dependence of the diffuse radiation on the solar
elevation angle.

## 6.2   Uncertainty of the radiation correction

The following components contribute to the overall uncertainty estimate $u(\Delta T_{\rm rad})$ for the radiation correction (note that '$\Delta$' denotes differences, not uncertainties):

1. Uncertainties from the laboratory experiments $u(\Delta T_{\rm exp})$ (Sect. 4.3 and Table 3), and uncertainties from radiation mod-
elling, $u(I_{\rm RTM})$ (Eq. 12). They were combined as:

$$u(\Delta T_{\rm exp,RTM}) = \Delta T_{\rm exp,RTM} \cdot \sqrt{\left(\frac{u(\Delta T_{\rm exp})}{\Delta T_{\rm exp}}\right)^2 + \left(\frac{u(I_{\rm RTM})}{I_{\rm RTM}}\right)^2}. \tag{18}$$





2. Uncertainties of pressure and ventilation in the actual radiosonde profile, converted to temperature uncertainties:

$$u(\Delta T_p) = \left| \frac{\partial(\Delta T)}{\partial(p)} \right| \cdot u(p), \quad u(\Delta T_v) = \left| \frac{\partial(\Delta T)}{\partial(v)} \right| \cdot u(v), \tag{19}$$

where the sensitivities are the partial derivatives of Eq. (5) with respect to $p$ and $v$, respectively.

3. Combination of the components in Eqs. (18) and (19):

$$u(\Delta T_{\text{dir,dif}}) = \sqrt{u^2(\Delta T_p) + u^2(\Delta T_v) + u^2(\Delta T_{\text{Exp,RTM}})}. \tag{20}$$

The subscript "dir,dif" in Eq. (20) indicates that the preceding steps were carried out equivalently for both the direct and diffuse radiation components $I_{\text{dir}}$ and $I_{\text{dif}}$.

The overall uncertainty of the radiation correction (Eq. 16) is then the combination of these two:

$$u(\Delta T_{\text{rad}}) = \sqrt{u^2(\Delta T_{\text{dir}}) + u^2(\Delta T_{\text{dif}}) + 2 \cdot \frac{\partial(\Delta T_{\text{dir}})}{\partial I_{\text{dir}}} u(I_{\text{dir}}) \cdot \frac{\partial(\Delta T_{\text{dif}})}{\partial I_{\text{dif}}} u(I_{\text{dif}})}. \tag{21}$$

It is assumed that the uncertainties of the direct and diffuse RTM results are strongly correlated because the occurrence of diffuse short-wave radiation is naturally linked to that of the direct light from the Sun. This is considered in Eq. (21) with a correlation term (GUM, 2008), assuming a correlation coefficient of $r = 1$. The sensitivities in that term are derived from Eq. (14) and (15) as

$$\frac{\partial(\Delta T_{\text{dir}})}{\partial I_{\text{dir}}} = \frac{\Delta T_{\text{dir}}}{I_{\text{dir}}}, \quad \frac{\partial(\Delta T_{\text{dif}})}{\partial I_{\text{dif}}} = \frac{\Delta T_{\text{dif}}}{I_{\text{dif}}}. \tag{22}$$

## 7  Comparison with manufacturer correction

To evaluate the GRUAN data processing, the GRUAN data product (GDP) for temperature is compared to the Vaisala data product (EDT) for a selection of 154 daytime and 81 nighttime soundings that were performed at Lindenberg observatory between 2014 and 2021. For each sounding, the differences between both products were gridded in 1000 m wide altitude

bins, that were used to collate the statistically averaged difference profile for all soundings. In the analysis the measurement uncertainties of the data were taken into account. The GRUAN-processed data contain for each data point an estimate of the uncorrelated uncertainty together with the correlated uncertainties in the spatial and temporal domains, where the latter two include the constant contribution from the calibration-uncertainty of the temperature sensor. This latter contribution is not relevant for the comparison of the radiation temperature correction of the same sensor by two different algorithms, and

leads to an overestimation of the resulting uncertainty. Therefore, for this analysis only the correlated uncertainties related to the radiation correction as well as the uncorrelated uncertainties were used. The uncorrelated uncertainties of the profile data averaged over a 1000 m altitude bin are added using the standard uncertainty of the mean:

$$u_{\text{uncorr}}(\overline{T}) = \frac{1}{N} \sqrt{\sum_{i=1}^{N} u_{i,\text{uncorr}}^2(T_i)}, \tag{23}$$





and the correlated uncertainty for the altitude bin is given by the mean of the correlated uncertainties:

$$u_{\mathrm{corr}}(\overline{\Delta T_{\mathrm{rad}}}) = \frac{1}{N} \sum_{i=1}^{N} u_{i,\mathrm{corr}}(\Delta T_{\mathrm{rad},i}). \tag{24}$$

The total uncertainty for each altitude bin follows from the geometric addition of the constituting components:

$$u(T_{\mathrm{GDP}} - T_{\mathrm{EDT}}) = \sqrt{u_{\mathrm{uncorr,GDP}}^2(\overline{T}) + u_{\mathrm{corr,GDP}}^2(\overline{\Delta T_{\mathrm{rad}}}) + u_{\mathrm{uncorr,EDT}}^2(\overline{T}) + u_{\mathrm{corr,EDT}}^2(\overline{\Delta T_{\mathrm{rad}}})}, \tag{25}$$

where the altitude-dependent correlated uncertainties of the Vaisala product $u_{\mathrm{corr,EDT}}(\overline{\Delta T_{\mathrm{rad}}})$ are less than $0.2\,\mathrm{K}$ ($k=1$) at $35\,\mathrm{km}$ (Vaisala, 2013; Jauhiainen, 2021). Due to the large number of data points $N$, $u_{\mathrm{uncorr}}(\overline{T})$ vanishes, so that the resulting total uncertainty reduces to $u_{\mathrm{corr}}(\overline{\Delta T_{\mathrm{rad}}})$.

The plot in Fig. 20 (a) shows a good correspondence between the nighttime temperatures over the entire profile. In the troposphere the GRUAN profile is slightly warmer than Vaisala, with the median of the difference smaller than $0.05\,\mathrm{K}$. Since neither GRUAN nor Vaisala processing do apply a radiation correction for nighttime soundings, the observed difference is attributed to the time-lag correction that is applied in the Vaisala processing. As a consequence of the absent radiation correction for the nighttime measurements, the resulting uncertainties also reduce to zero, and are therefore not visible in the left panel of Fig. 20.

The plot in Fig. 20 (b) shows that GRUAN-processed daytime profiles are up to $0.1\,\mathrm{K}$ warmer in the troposphere, and above the tropopause (approximately $10\,\mathrm{km}$) the temperature difference steadily increases, with the GRUAN profile $0.35\,\mathrm{K}$ warmer at $35\,\mathrm{km}$. The observed daytime differences between the GRUAN and Vaisala-processed data are within their combined uncertainty-range for $k=2$:

$$|T_{\mathrm{GDP}} - T_{\mathrm{EDT}}| < k \cdot \sqrt{u^2(\overline{\Delta T_{\mathrm{rad,GDP}}}) + u^2(\overline{\Delta T_{\mathrm{rad,EDT}}})}, \quad (k=2), \tag{26}$$

meaning that according to the classification scheme proposed by Immler et al. (2010), both data products are, albeit barely, in agreement.

## 8 Summary and conclusions

The paper presents SISTER, a setup that was developed to quantify the solar heating of the temperature sensor of radiosondes under laboratory conditions by recreating as closely as possible the atmospheric and illumination conditions that are encountered during a daytime radiosounding ascent. The resulting parameterisation of the radiation error as a function of pressure, ventilation speed and actinic flux is applied in the GRUAN processing for radiosonde data, which relies on the extensive characterisation of sensor properties to produce a traceable reference data product which is free of manufacturer dependent effects. The GRUAN radiation correction model combines the laboratory characterisation with model calculations of the actual radiation field during the sounding to estimate the correction profile. The second part of the paper describes how this procedure was applied in the development of the GRUAN data product for the Vaisala RS41 radiosonde (version 1, RS41-GDP.1).

(a)

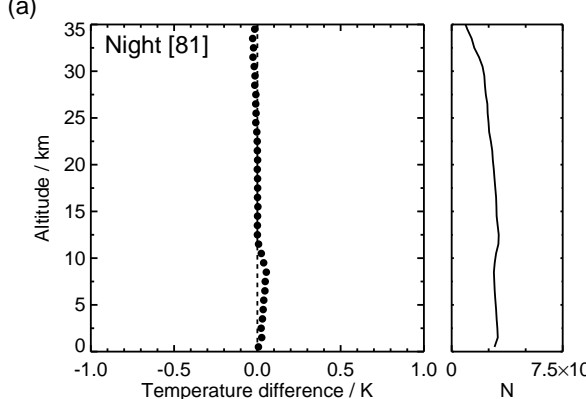

(b)

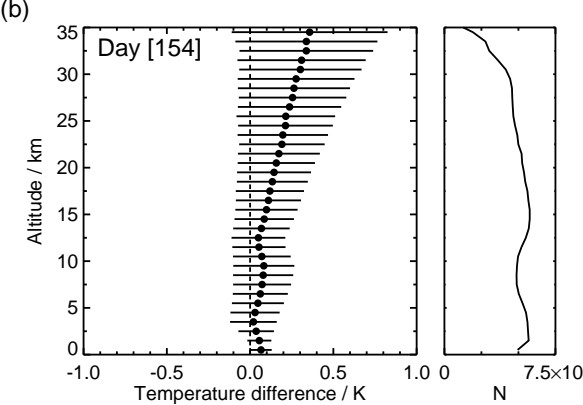

**Figure 20.** Comparison of GRUAN data product (GDP) and Vaisala product (EDT) for RS41 temperature radiosoundings performed in Lindenberg between 2014 and 2021. Data were taken from radiosoundings performed at nighttime (left, solar elevation angle $\alpha_s < -10°$, $N = 81$) and daytime (right, $\alpha_s > 10°$, $N = 154$). The main graphs show the median of $T_{\mathrm{GDP}} - T_{\mathrm{EDT}}$ (solid dots) versus altitude for 1000 m wide altitude bins, and the side plots show the number of data points in each altitude bin. The horizontal bars represent the uncertainty ($k = 2$) that is composed of the correlated and uncorrelated uncertainties of both data products. The correlated uncertainties for nighttime measurements are absent, because no radiation correction is applied.

SISTER controls the pressure (3 hPa to 1020 hPa) and ventilation speed ($0\,\mathrm{m\,s^{-1}}$ to $7\,\mathrm{m\,s^{-1}}$) of the air inside the setup to simulate the conditions between the surface and 35 km altitude, to determine the dependence of the radiation temperature error on the irradiance and the convective cooling. The radiosonde is mounted inside a quartz tube with the sensor boom unfolded in a flight-like configuration, while a 20 cm wide beam from an external 2500 W light source illuminates the complete sensor boom of the radiosonde to include the conductive heat transfer between sensor and boom. A special feature of SISTER is that the radiosonde is rotated around its axis to imitate the spinning of the radiosonde in flight. The averaging due to this rotation yields the sensor heating that depends on the effective exposed surface of the sensor boom. The irradiance geometry can be adjusted to accommodate solar elevation angles 0° to 60°, and, in addition, 90°. A special configuration was used to quantify the heating due to diffuse radiation. LDA characterisation of the airflow inside the quartz tube showed a predominantly laminar flow in the longitudinal direction, with a velocity profile typical for a tubular flow. The LDA measurements were also used to calibrate the speed of the airflow at the position of the radiosonde between 20 hPa and 1000 hPa. Measurements are performed for various pressures, ventilation speeds and illumination angles, yielding a 2D-parameterisation of the radiation error for each illumination angle. Fig. 13 shows that the uncertainty of this parameterisation is less than 0.2 K for typical ventilation speeds ($3\,\mathrm{m\,s^{-1}}$ to $7\,\mathrm{m\,s^{-1}}$).

The GRUAN data processor calculates the direct and diffuse short wave radiation profiles using the embedded Streamer RTM, where the temperature and humidity profiles are taken from the actual sounding data. By lack of accurate cloud in-





formation, a generic cloud scenario is used, which increases the uncertainty of the radiation profile. The averaged correction

profile increases gradually from 0.1 K at the surface to approximately 0.8 K at 35 km altitude. Comparison between sounding data that were GRUAN-processed (GDP) and Vaisala-processed (EDT) reveal no differences for nighttime soundings, where the radiation correction is not applied, apart from a small offset in the troposphere that is attributed to the time-lag correction applied by Vaisala. The daytime differences are smaller than $+0.1$ K (GDP-EDT) in the troposphere and increase above the tropopause steadily with altitude to $+0.35$ K (GDP$-$EDT) at 35 km. These values are just within the limits of the combined

$k = 2$ uncertainties of both data products, which means that the GRUAN processing and the Vaisala processing are in agreement. It is not possible to determine which of both data products performs better, although the GDP represents the best-effort for the characterisation and correction of the radiation temperature error. This unresolved finding exemplifies the need for an independent reference instrument for in situ temperature measurements.

In its current form the correction algorithm uses a standardised cloud scenario in the estimation of the radiation profile, which

as a result has a considerable uncertainty, especially in the troposphere. The accuracy of the estimated radiation profile could be greatly improved if real-time information on the actual cloud configuration during the sounding were available. A potential source for in-flight cloud information is the measured humidity profile. For air temperatures above freezing, 100 % humidity indicates the presence of wet-phase clouds. However, at lower temperatures this approach has its limitations because the presence of ice-phase clouds can not unambiguously be established from the humidity profile. Furthermore, the humidity profile

provides in situ information on cloud presence, which would be sufficient for a homogeneous cloud cover, but insufficient in case of scattered clouds, or if the cloud scenario changes along the balloon's trajectory. The latter could be resolved by using additional cloud information from an external source, such as satellite observations. The incorporation of space-borne cloud information would require the collection of the appropriate data covering the time of each sounding and their integration into the processing, which is a complicated effort. Still, such an extension is considered for a future version of the GRUAN

processing, which despite its complexity is expected to considerably improve the accuracy of the radiation correction.

The SISTER setup is designed for measurements at room temperature, and contrary to, e.g., the Upper Air Simulator described by Lee et al. (2018a), it does not have the possibility to cool down the radiosonde or the chamber to simulate the conditions at the tropopause. Because of the $T^4$-dependence of the emitted power, the energy transfer by long wave radiation at $-60\,°C$ is approximately four times smaller than at room temperature, and this reduced cooling efficiency at low temper-

atures does in principle increase the radiation temperature error. However, it is not expected that this will affect our findings because the magnitude of the heat exchange by long wave radiation is, due to the metallic coating of the sensor boom, considerably smaller than the other contributions to the sensor's heat budget, such as the shortwave radiative heating and the convective cooling (Luers and Eskridge, 1995).

Fig. 18 shows that vertical air movements such as gravity waves can cause disparities between the GPS-based ascent speed

and the actual speed of the balloon rising through the surrounding air, which in turn introduces oscillations in the temperature correction profile. These oscillations can lead to a temporary under- or over-correction of the radiation temperature error, which affects the quality of the data product. A possible way to remedy this is by adopting the method proposed by Wang et al. (2009) for quantifying vertical air movements using still-air balloon rise rates.



Fig. 11 shows that changes in the exposed area due to the rotation of the illuminated radiosonde introduce oscillations in the temperature measurements. The analysis of the laboratory experiments (Sect. 4.1) shows that the determination of the resulting radiation error is not affected by this rotation, but that these oscillations are well resolved due to the fast response time of the temperature sensor and sensor boom system. This finding can be exploited to improve the radiation correction: using information on the actual orientation of the radiosonde from, e.g., an additional, position-sensitive sensor, will lead to a better estimate of the instantaneous radiation temperature error.

SISTER is designed to characterise modern radiosonde types from various manufacturers. As a consequence, it is intended to use this setup to support the development of GRUAN data products for other sonde models. First measurements with other radiosonde models have already been conducted. Consistency checks by comparing the RS41 GDP temperatures with those from (future) GDP's of other radiosondes will be an important quality test for the presented method.

*Data availability.* The data from the laboratory experiments that were used for Figs. 11–13, a list to data sources for Figs. 19 and 20, and two image files showing photography on how the radiosonde is installed, are stored as zip-file in a permanent repository at https://www.gruan.org/data/data-packages/dpkg-2021-1. Further data are available upon request.

*Author contributions.* CvR scientifically supported the development of the experimental setup, guided and executed the measurements, analysed the data, and prepared essential parts of the manuscript. MS adapted the radiation model, implemented the radiation correction procedure in the GRUAN RS41 data processor, and prepared the related sections for the manuscript. TN initiated and lead the technical development and construction of the experimental setup. VM prepared and conducted the LDA measurements. RD performed the comparison analysis, and made essential contributions throughout the manuscript.

*Competing interests.* The authors declare that no competing interests are present.

*Acknowledgements.* The authors wish to thank the technicians at MOL-RAO (R. Tietz and H. Friedrich) for their support with setting up and installing the system and carrying out the measurements. We also thank C. Egbers (Chair of Aerodynamics and Fluid Mechanics, Brandenburg University of Technology Cottbus-Senftenberg) for cooperation with regard to the LDA measurements.





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
