# Peer review of "Laboratory characterisation of the radiation temperature error of radiosondes and its application to the GRUAN data processing for the Vaisala RS41"

_Atmospheric Measurement Techniques, 2021_

## Author Comment (AC1)

**Response to Reviewer #1 (RC1)**

We want to thank the reviewer for his/her valuable comments and suggestions that will help to improve the quality of the manuscript.

Major comments:

"The GRUAN product for the Vaisala RS92 temperature measurements was established using a predecessor of this new and improved radiation simulator. The GRUAN product for the Vaisala RS41 is based on the measurements using the current simulator. The authors did not mention this transition and how the radiation error correction based on these new
measurements might compare to the radiation error correction measurements of the RS92 using the older simulator. Ideally, the Vaisala RS92 would have been evaluated using SISTER as well, but given the complexity of these measurements, this may not have happened. Nevertheless, some estimation of the difference between the RS92 corrections using the older and the RS41 corrections using the newer simulator is needed to gauge what
systematic error this new simulator might introduce in the measurements."

We agree that a comparison of the temperature radiation effect for the RS92 using the old and the new setup, together with a comparison of the RS41 results in the new setup, are necessary to evaluate the quality of the existing RS92 radiation correction derived in the old setup, as well as the performance of SISTER. Measurements with the RS92 in
the new set- up have not been performed yet, mainly because of the urgency to develop a GRUAN data product for the RS41 after the near network-wide transition from RS92 to RS41 as operational radiosonde within GRUAN. The main focus of this paper is to present the SISTER setup and to describe the GRUAN radiation correction for the RS41 derived from measurements with SISTER. An additional comparison with the RS92
GRUAN data product is beyond the scope of this paper and would make it too long. A separate paper that includes an extensive comparison of the GRUAN data products for the RS41 and the RS92 is in preparation.

The SISTER setup is inspired by the previous setup, with substantial improvements to overcome the limitations of the latter. The GRUAN radiation correction for the RS41
relies on the analysis of an ensemble of radiation measurements, which includes, amongst others, the investigation of the influence of the sensor orientation. The assessment of differences between the radiation corrections derived from both setups, based on the comparison of a few measurements, seems not feasible due to the conceptual differences between SISTER and the old setup, and in the subsequent data
analysis.

"I disagree with the evaluation that the radiation correction based on SISTER is statistically consistent with that provided by Vaisala. While this is true for each individual profile, for the mean correction in a dataset of 154 sonde, the factor 1/sqrt(154) in Equation 20 should
substantially reduce the uncertainty estimate for the entire data set and make it statistically different from that of Vaisala.

While this is negligible for forecasting, this would be important for long term climate series. Of course, this does not answer which algorithm is correct; but, as the authors point out, the GRUAN approach is, at least, well documented and traceable, whereas that of Vaisala is
not."

The uncertainty estimates in the GRUAN data product include correlated and uncorrelated components. The uncorrelated components are evaluated with Eq. 23, and do indeed reduce by 1/sqrt(N). The correlated uncertainties on the other hand are evaluated with Eq. 24, and do not disappear with increasing N, see e.g. Immler (2010). The term "in agreement" means that the uncertainty of the difference is not smaller than the difference for a coverage factor of k=2.

"Section 5 is probably very similar to what was done by Dirksen et al. (2014); however, there is no reference to that paper in this section. Dirksen et al. (2014) is mentioned in the introduction, but here it would be good to highlight, what the differences are to that paper, e.g. the treatment of the zenith angle near the horizon and maybe some other aspects."

In the RS92-GDP, the RTM streamer was for the first time used to simulate the radiation. The simulations were carried out in advance for a few individual cases or grid points and saved in a Look-Up Table (LUT). During the actual processing, only this LUT was used, and linear interpolation was performed between the grid points. No data from the actual radiosonde profile such as temperature and humidity were used. The surface albedo was estimated with constant values.

In the current version for the RS41, there are substantial improvements with regard to the way how the RTM is used and connected with the processing. Key points are that the Streamer is integrated into the processing, i.e. radiation profiles are simulated by calling up the Streamer routines from within the processing in several runs for each individual sounding. The actual temperature and humidity profile is taken into account, as well as existing regionally and seasonally representative albedo information. The treatment of problematic solar zenith angles close to the horizon (Streamer assumes a flat surface) is significantly advanced. Also, the change of the "effective" horizon with the height of the sonde is taken into account.

In the introduction of Section 5 we will mention Dirksen et al. (2014) and highlight the differences between that paper and the current approach.

Minor comments:

"Section 2.2: Can you make a statement how far the setup is from turbulent flow, i.e. is there anywhere in the parameter space a risk that the laminar flow will change to turbulent flow?"

We cannot make a clear statement on this. The LDA data indicate that the axial component of the flow velocity exceeds the radial by an order of magnitude. We consider this to be sufficient indication that the in-flight flow conditions are reproduced well enough, and we assume that - under these conditions - a small amount of turbulence does not affect the measurement results.

"Line 285: The pressure sensors in the RS41 radiosondes usually have an offset, that is compensated for during the ground check. Was a similar pressure correction done here? The offset correction may easily be in the range 0.5 to 1.5 hPa for an individual sonde and would affect the low pressure analysis."

A ground check with the RI41 unit was performed prior to the measurements. It is worth mentioning here that the correction of the pressure sensor as employed by Vaisala is a scaling factor based on the pressure difference observed in the RI41 unit at surface pressure. Therefore, the error of the RS41's pressure sensor scales with pressure, meaning that an uncorrected calibration error of the p-sensor is negligible below say 20 hPa.

"Section 2.3.6: I understand the argument how to simulate the diffuse radiation with direct radiation. However, there should be a geometric scaling factor, which expresses the difference between the two. I assume this is hidden in the flux of 527 Wm$^{-2}$ that was used in this measurement. This scaling needs to be explained a little better."

      There is no scaling factor applied. As stated in lines 379-380, the heating by diffuse
radiation with a flux of 527 Wm$^{-2}$ is identical to the heating caused by a perpendicular incident beam of the same flux. For the 'diffuse' experiments, the sensor boom was irradiated perpendicularly and without rotation of the sensor boom. This imitates the constant irradiation by diffuse light in soundings, which is independent of the actual orientation of the sonde (during a sounding, both sides of the boom are warmed equally
by diffuse radiation). The value of 527 Wm-2 is arbitrarily selected (but similar to the level of diffuse radiation in the atmosphere) via the distance of the setup to the lamp. The measured 'diffuse' Delta T is linearly scaled before use as input for the radiation correction, equivalent to the 'direct' measurements.

"Lines 376-384: It is not clear what is explained here. Is this another explanation for section 2.3.6? Is the difference of the diffuse curves not an indication that the value of 527 Wm-2 is too large and that the geometric scaling factor should be something different? Please clarify."

      Indeed, these lines refer to section 2.3.6. After linear scaling to a common reference irradiance, which makes the results from the different setups comparable, the 'diffuse'
sensor warming should be larger than that from the 'direct' setups, because for the latter (at the same irradiance) the cosine effect takes effect due to sonde rotation and the non-perpendicular incidence angles that simulate different sun elevations. Lines 379-381 are supposed to say that.

"Line 350 and Equation 5: Is there any physical justification for the simple model? Why did the authors choose 1/sqrt(v) and 1/sqrt(p) in the polynomial? Isn't the deviation from that model at low pressures an indication that it may not be the most suitable fit?"

      The intention of the curves in Fig. 12 is to demonstrate that $\Delta T$ roughly follows an inverse square root dependency on $p$ and $v$. The curves do not represent final results
but are shown to motivate the use of a polynomial in $1/\sqrt{v}$ and $1/\sqrt{p}$ as fit model for the quantitative evaluation (Eq. 5). The physically evidenced motivation for this $1/\sqrt{v}$ and $1/\sqrt{p}$ parameterisation is given in the following:

      The heat exchange coefficient $h$ describes the rate of convective cooling of the irradiated sensor. It can be parameterised using the Nusselt number $Nu$:

$h(Nu) = Nu \cdot \lambda/l$ (definition of Nusselt number), with $l$ the object (sensor) dimension, and $\lambda$ the thermal conductivity of air.

      $Nu$ is estimated from Reynold's number $Re$ (see e.g. Luers (1990), for a cylinder in a laminar cross flow):

$$Nu = 0.184 + 0.324 \cdot Re^{0.5} + 0.291 \cdot Re^{(0.247 + 0.0407 \cdot Re^{-0.168})}.$$

The $Re^{0.5}$- term is essential here, the last term contributes significantly less.

$Re(v, p)$ is defined as

$$Re(p, v) = \frac{vl\rho}{\eta} = \frac{vl\rho_0}{\eta} \cdot \frac{p}{p_0},$$

with $l$ the object (sensor) dimension, $\lambda$ the thermal conductivity of air, $\rho_0$ and $p_0$ the density and pressure of air at normal conditions, $\eta$ the dynamic viscosity. In a first approximation:

$h(Nu(Re)) \sim Re^{0.5}$, and thus $h \sim p^{0.5}$ and $h \sim v^{0.5}$.

It can be assumed that $\Delta T$ is inversely proportional to $h$, i.e. the stronger the convective cooling of the sensor, the smaller the $T$-effect due to radiation. Therefore:

$\Delta T \sim 1/(p^{0.5} v^{0.5})$, which motivates our approach for the parameterisation of the 2D-fit.

"Figures 4 and 13 are pretty to look at, but not very helpful in evaluating quantitative differences. I can't tell exactly, where different data points belong. The left panel of Figure 13 has the corresponding line plot in Figure 14, but Figure 4 and the right panel of Figure 13 do not. I would suggest replacing both with a suitable line plot."

Strictly speaking, x-y-plots of the measurement points cannot directly be created because the grid of pressure and ventilation settings is not always completely equidistant or regular. For this reason, the points in Fig. 12 e.g. are binned (see legend). The lines in Fig. 14 represent the measurement data in a 'pre-evaluated' form as curves based on the 2D-fits to the measurement points, linearly up-scaled to a common fixed irradiation value. Fig. 13 is thought to give a first qualitative overview over the general distribution of the raw Delta T as well as the estimated uncertainties over the p-v-space, whereas Fig. 14 gives a more quantitative example. Similar applies to Fig. 4. We prefer a few rather intuitive representations as a first impression at a glance over creating numerous plots for exact quantitative reading, which we think to have limited value for general understanding.

"I could not follow the discussion of the uncertainty interpolation in lines 402-412. Please rewrite."

The paragraph will be modified to improve clarity.

"The right panel of Figure 13 as line plot may be a big help in explaining what is happening here."

See reply to the second last point above

"Lines 604ff: Vaisala uses a time lag correction for the temperature, GRUAN does not. At Lindenberg this seems to be justified. However, at tropical stations, which have a strong temperature gradient also in the stratosphere, this may have a stronger effect. Have you looked at that? Does the Lindenberg result still hold in the tropics?"

This paper solely focuses on the characterisation and correction of the radiation error of the temperature sensor, therefore we did not perform independent measurements of the time-lag for the RS41 temperature sensor. According to the specifications provided by Vaisala, the time lag of the T-sensor is 0.5 s at 1000 hPa, and although it will increase at lower pressures, it still will be a challenge to accurately determine the lag in our laboratory setups, considering the 1s-sampling by the radiosonde. Therefore, a correction for the time lag of the temperature sensor is not implemented in the GRUAN data processing for the RS41. A time lag correction based on the value provided by Vaisala has not been considered yet.

The comparison of the night time temperature profiles in Figure 20 do indeed show no differences in the stratosphere over Lindenberg that can be attributed to the time lag of the temperature sensor, which is consistent with the absence of significant temperature gradients in the local stratosphere. A preliminary analysis of profiles from tropical sites indicate that for these sites time lag-related differences are visible in the stratosphere, but that the magnitude is smaller than in the troposphere.

Technical comments:

Line 18: Comparisons will be changed as suggested

Line 27: Delete "for example"

will be deleted as suggested

Line 54: Change "reduces" to "decreases"

will be changed as suggested

Line 55: delete "the decreasing"

will be changed as suggested

Line 75: "... by direct ..."

will be changed as suggested

Line 77: change "caused by" to "due to"

will be changed as suggested

Line 83: "Following the GRUAN ..."

will be changed as suggested

Line 87: "... radiative flux ..."

We prefer to keep the phrase 'actinic flux', since it is the absorption of the actinic flux (the spherically integrated radiation flux in the earth's atmosphere that originates from the sun, including the direct beam and any scattered components) that causes the heating of the temperature sensor.

Line 88: Change "By lack ..." to "Due to the lack ..."

will be changed as suggested

Line 90: Change "... solar position ..." to "... position of the sun ..."

will be changed as suggested

Line 94: "... applied to the ..."

'Applied' will be replaced by 'employed'. With this sentence we want to express that the complete GRUAN approach (starting at line 83: laboratory characterisation, parameterisation of the radiation error, estimation of the radiation profile, correction) is employed in the process of developing the GRUAN data product, and that not only
certain aspects were used in the GRUAN data processing for the RS92.

Line 103: "the Lindenberg Observatory ..."

will be changed as suggested

Line 104: change " ... of the SISTER ..." to " ... of SISTER ..."

will be changed as suggested

Line 105: "... an unfolded ..."

will be changed as suggested

Line 106/107: change "together with" to "and"

will be changed as suggested

Line 108: "... and includes ..."

Will be changed as suggested

Line 121: LDA has not yet been spelled out

We will insert 'Laser Doppler Anemometry' before using the abbreviation LDA

Line 140: change "one of the middle legs" to "one leg"

will be changed as suggested

Line 140: Does 180 mm refer to the diameter of a round tube or the width of a rectangular tube. Figure 1 and the description are a little fuzzy on this point.

mm is the diameter of the cylindrical quartz tube. The metal casing that is visible in Fig. 2 is a safety cover. We will add a sentence clarifying this.

Line 141: Change to "... is mounted. To generate a radially uniform flow a rectifier ..."

Will be changed as suggested. A sentence mentioning the strainers for suppression of turbulence is added.

Line 150: change "radiosonde's casing" to "housing of the radiosonde electronics"

Will be changed as suggested

Line 153: I can't really see the threads in the Figures. Maybe just remove the reference to
Figure 1 and 2.

References to Figures will be repositioned within the sentence.

Line 166: "... an RS41 ..."

Will be changed as suggested

Line 168: delete "which is"

Will be changed as suggested

Line 172: delete "generally speaking"

Will be changed as suggested

Line 174: "flow velocity profiles"

'velocity' will be inserted

Line 180: delete "which is"

Will be changed as suggested

Line 180: How did you extrapolate to pressures below 20 hPa?

The extrapolation is provided with the fit in Eq. (1). We estimate the 1-sigma uncertainty of the ventilation speed to be 0.5 m/s as a constant value. The deviations of the measured data points from the fit (Fig. 4) are well within that limit, and we assume that the limit also covers uncertainties connected to the extrapolation to the pressure range below 20 hPa.

Line 206: The lamp flux decreases with distance, not the lamp output.

Will be changed as suggested

Line 210: Units should probably be Wm-2

The units in Eq. (2) are Wm-2 for the flux I(r), and as stated in the text, W for the fit parameter P0, and m for the distance r and the other fit parameter r0.

Line 238: "... sensor boom ..."

Will be changed as suggested

Line 258: " ... typically for ..."

Will be changed as suggested

Line 260: Delete "again"

Will be deleted

Line 266: Delete "that are"

Will be deleted

Line 268: change "for various" to "at different"

Will be changed as suggested

Line 272: Just a comment: It might have been good to replace the incidence angle of 20 deg with 75 deg.

We agree that this would be preferable. However it was not possible to perform measurements for incidence angles between 60° and 89° due to limited space on the optical bench when turning the setup in front of the fixed lamp to set the angle.

> The measurements at 90° incidence were performed in a special configuration of the setup, as described in the manuscript.

Line 294: delete "rapid"

> Will be deleted

Line 296: delete "Temporal"

> Will be deleted

Line 300: delete "temporally-consistent"

> Will be deleted

Line 325: spaces before and after "and"

> Will be corrected

Line 326: "temperature, "

> Will be corrected

Line 346: delete "also in a complex way"; change "turbulence conditions" to "turbulent flow"; change "are" to "is"

> Will be modified as suggested

Line 350: Delete "It is found that"

> Will be deleted

Line 358: change "monotonous" to "monotonic"

> Will be changed

Line 365 ff: better "The fits were created for all of the six incident angles, i.e. for 0 deg, 20 deg, 40 deg, 59 deg, zenith and diffuse. The two zenith and the two diffuse radiation configurations were averaged as explained in Sect. 2.3.5 and 2.3.6"

> Sentences will be rephrased. We prefer to say that we have five incidence angles and a diffuse configuration. The assignment of an angle for the simulation of diffuse radiation may be somewhat counterintuitive, although in practice direct radiation at a specific angle was used. The sentence will be modified in that sense.

Line 367f: Delete sentence "The parameterization ...". The reference is from that section to
Eq 5.

> Will be deleted as suggested

Line 369: Hasn't the normalization to a constant irradiation not already been done? Why mention this here?

> Up to this point, the normalisation is not yet done. Coefficients according to the
> parameterisation with Eq. (5) are determined individually for the measurement results from the different illumination configurations (with individual irradiances). Linear scaling ('normalisation') to a common arbitrary irradiance is then used for each of them to enable direct comparison of the measurement results. Such a comparison is demonstrated in Fig. 14. The caption of Fig. 14 as well as the sentence around line 370
> will be modified for better understanding.

Line 386: change "leaps" to "steps"

Will be changed as suggested

Line 397: Delete, this seems to be a repeat of the previous explanation.

From the right panel in Fig. 12 and using the slopes of the curves one can read that the uncertainty of the ventilation (0.5 m/s) - translated into an uncertainty component for Delta T -  dominates the other components, which might not be obvious from the list with the four points. The sentence should point this out. We will change 'Ventilation speed...' in line 397 to 'The uncertainty of ventilation speed …'.

Line 400: What is "plus another component"? Do you refer to the factor 1/(2 * SQRT(3))? If so, you could reference GUM and point out that the value lies with equal probability somewhere in that range.

The sentence will be rephrased.

Line 415: A set of lines plot would be better to show this difference.

See earlier replies

Lines 433 and 446: Change "on" to "onto"

Will be corrected

Line 454: change "augmented" to "influenced"

Will be changed as suggested

Equation 7: change "=" to "≈"

Will be changed.

Line 541: "The apparent discontinuities ..."

'apparent ' will be inserted.

Lines 545ff: Gravity waves can happen anywhere above the tropopause, i.e. also below 25 km. The question is, shouldn't you use a form of theoretical rise rate instead of the measured to avoid biasing the temperature profile? You discuss this in the Summary and Conclusions, without actually reaching a conclusion. Maybe delete this short discussion here and add a sentence or two in the Summary.

The altitude of 25 km is not meant as a general statement on the occurrence of gravity waves, but refers to the specific case that is displayed in Fig. 18. We will adjust the text accordingly to reflect this. In the Summary and Conclusions we discuss this issue because it needs to be addressed and improved in a future version of the data processing. A potential way of solving this is, as indicated, applying the method by Wang et al.

Line 554: Why did you only use 154 profiles and not years' worth of profiles (several thousand?)

The GRUAN data processing for the RS41 radiosonde, in which the radiation correction presented in this paper is implemented, has just been completed and the data product will be officially released soon. The processing of the existing GRUAN data archive for RS41 soundings is ongoing, which will take some time. A more detailed statistical analysis, taking into account more profiles, but also differentiating by e.g. climate zone and season, will be the subject of a separate study and paper. The presented analysis for 154 soundings is intended to give a first general impression on the magnitude and variability for one site.

Figure 20: The right panels seems to have squeezed vertical axis labels.

The left and the right panels have a common vertical axis, therefore the axis labels are omitted in the right panels. We will make the size of the ticks in the left and the right panels identical.

Line 623: "... and the ventilation speed ..."

'the' will be inserted

Line 635: Delete: "Fig 13. shows that"

Deleted as suggested

Lines 654ff: Wouldn't a cloud model based on radiosonde RH still beat the dumb statistical assumption in most cases?

We have not tried to assess whether the inclusion of the RH profile from the radiosonde
data would on average result in a more representative cloud estimate. This is difficult to evaluate without comparing with information from other methods (such as satellites etc.). It seems especially difficult to estimate the uncertainties that are connected to such an approach, which are not necessarily smaller, but more properly defined when using the mean of two 'extreme' scenarios as in this study.

Line 674: Delete "Fig. 11 shows that"

Will be deleted as suggested

---

## Author Comment (AC2)

**Response to Reviewer #2 (RC2)**

**Comment:**

"One major concern is that the SISTER setup operates only at room temperature. As previously reported, the radiation correction of the RS41 temperature sensor presented a temperature dependency (Lee et al. Meteorol. Appl. 27, e1855, 2020). In the paper, the radiation correction value (delta T) of the RS41 is increased by 18% as the temperature is decreased from 20 °C to -70 °C at a constant p = 7 hPa and v = 5 m/s. In this regard, the low temperature effect cannot be ignored by the sentence "However, it is not expected … and the convective cooling." in lines 665-668 in Summary and conclusions."

**Reply:**

We are aware of the measurements presented in Lee (2020). Indeed, the measurement points in Fig. 9 of that paper suggest an increase of $\Delta T$ at the lowest temperatures at least for pressures below 30 hPa. However, as noted in the conclusions of Lee 2020, no uncertainties were evaluated for the $\Delta T$-measurements, so that we think a reliable assessment of the results is not (yet) possible at this stage. We think that more information is required, e.g. experimental data or quantitative model calculations, to evaluate if and to what extent the dependence on the ambient temperature is significant, and to clarify whether the observations can be explained primarily by long-wave radiation. We are looking forward to learning more in the upcoming study amt-2021-246 (see reply to next comment below).

The sensitivity of convective cooling to decreasing absolute temperature is not directly obvious, since both thermal conductivity and viscosity decrease, whereas density increases. As Lee et al. state, the absolute values for $\Delta T$ will be lower at a more realistic incoming flow with inclined sensor boom (because of higher flow resistance and more efficient heat loss), but the sensitivity to absolute temperature may be different as well. It is not straightforward to evaluate whether the convective cooling at low pressure loses significance such that longwave cooling dominates when temperature decreases.

There are significant differences between the long wave radiation environment in the laboratory setup and in the free atmosphere. In the laboratory setup there is a uniform background emitted by the walls of the measurement chamber, with only a small temperature difference with respect to the temperature sensor. In the free atmosphere the long wave radiation background is composed of, amongst others, contributions from the air masses and surface below and from the cold cosmic background. Therefore, it cannot be excluded that the observed low-temperature effect is to some extent specific for the conditions inside the measurement chamber.

**Comment:**

"Recently, Lee et al. submitted a new paper to the Atmospheric Measurement Techniques (amt-2021-246) which deals with a potential solution for this issue by providing a formula to estimate delta T at low temperatures by only using measurements at room temperature. In the paper, delta T of the RS41 is increased by 20% as the temperature is decreased from 20 °C to −67 °C at a constant p = 5 hPa and v = 5 m/s. Therefore, it is more desirable for the

GRUAN to include the low temperature effect for the radiation correction for the GRUAN data processing. I would suggest the authors to comment on this point in their Conclusions."

**Reply:**

We are very much looking forward to reading the results presented in amt-2021-246 once it is published. As the paper in question was not available when we prepared our manuscript (and at this moment still is not available), it is not possible to include a discussion of it in our manuscript.

Specific comments:

1) Line 665-668: The sentence "However, it is not expected ... and the convective cooling." should be revised because the low temperature effect on the RS41 temperature sensor was observed previously and again recently (amt-2021-246). The same phenomenon was also observed for thermistor-type T-sensors even though there is no apparent air ventilation (Lee et al. Meteorol. Appl. 25, 283, 2018). Based on the fact that the low temperature effect appeared when the convective cooling was limited (no apparent ventilation), it is likely due to the T4 dependent long-wave radiation from the sensor.

**Reply:**

We will revise the relevant paragraph and add a reference to Lee (2018).

---

## Referee Report (RR1)

**Comment to Authors:**

Although following comment can partially be an answer to the GRUAN's comments to our paper (amt-2021-246), the issue on the mechanism behind the temperature effect is also important to this paper (amt-2021-187).

We have consulted with an expert on heat transfer and found that the temperature effect is mainly because the thermal conductivity of air is decreased as the air temperature is lowered (**Fig. 1**). We have also learned that long-wave radiation from the sensor is negligible whether the sensor temperature is 20 °C or -70 °C as originally insisted by the authors.

[Figure]

**Figure 1**. Thermal conductivity of $N_2$ gas at 7 hPa as a function of temperature.

To make this matter as simple as a textbook example, a metal sphere is considered to be a temperature sensor under solar irradiation with varied temperature as shown in Fig. 2.

[Figure]

**Figure 2**. Simplified example to show the temperature effect on $T_s$ - $T_a$.

Equation governing the heat transfer of the sphere is as follows:

$$q = h \cdot A(T_s - T_a) \tag{1}$$

, where $q$ is heat flow (W) due to solar irradiation, $h$ is heat transfer coefficient (W/($m^2$K)), $A$ is surface area ($m^2$), and $T_s$ and $T_a$ are the sensor and air temperature (K), respectively. Long-wave radiation from the sensor is not considered in Eq. (1) since it is negligible. Thus, the heat

transfer from the sensor is governed by $h$ which is a function of various properties of air including density $\rho$ (**Fig. 3**), ventilation speed (fixed at $v = 5$ ms$^{-1}$), viscosity $\mu$ (**Fig. 4**), heat capacity $C_\mathrm{p}$ (**Fig. 5**), and thermal conductivity $k$ (**Fig. 1**).

[Figure]

**Figure 3**. Density of $N_2$ gas at 7 hPa as a function of temperature.

[Figure]

**Figure 4**. Viscosity of $N_2$ gas at 7 hPa as a function of temperature.

[Figure]

**Figure 5**. Heat capacity of $N_2$ gas at 7 hPa as a function of temperature.

**Figure 1, 3, 4, and 5** are adopted from NIST Chemistry WebBook in the following address:

https://webbook.nist.gov/cgi/fluid.cgi?P=0.0007&TLow=-100&THigh=50&TInc=1&Applet=on&Digits=5&ID=C7727379&Action=Load&Type=IsoBar&TUnit=C&PUnit=MPa&DUnit=kg%2Fm3&HUnit=kJ%2Fkg&WUnit=m%2Fs&VisUnit=uPa*s&STUnit=N%2Fm&RefState=DEF

Using these properties of air (all known), the heat transfer coefficient $h$ in this spherical geometry is calculated based on a textbook equation as follows:

$$h = \frac{k}{D} \times \left[ 2 + \left( 0.4 Re^{\frac{1}{2}} + 0.06 Re^{\frac{2}{3}} \right) \cdot \left( \frac{\mu C_p}{k} \right)^{\frac{2}{5}} \right] \qquad (2)$$

, where $D$ is diameter of the sphere and $Re$ is the Reynolds number calculated by $\rho v D / \mu$. Eq. (2) is adopted from "Introduction to Heat Transfer" by F. P. Incropera & D. P. DeWitt, Fourth Ed. Chapter 7.5.

Parameters and their values used for the calculation of Eq. (2) is summarized in **Table 1**.

The heat transfer coefficient $h$ is reduced by about 20% when air temperature ($T_a$) is varied from 20 °C to −70 °C at a fixed pressure of 7 hPa and ventilation speed of 5 m·s⁻¹. Consequently, the radiation correction value ($T_s$ - $T_a$) at $T_a = -70$ °C is increased by about 20%. This number is surprisingly in good agreement with the experimental result by the upper air simulator (UAS).

**Table 1**. Parameters and values for calculation of $h$ and ($T_s$ - $T_a$).

| Parameter | Symbol (Unit) | Value ($T_a$ = 20 °C) | Value ($T_a$ = −70 °C) |
|---|---|---|---|
| Diameter | $D$ (m) | 0.001 | 0.001 |
| Air pressure | $P_a$ (Pa) | 700 | 700 |
| Ventilation speed | $v$ (ms⁻¹) | 5 | 5 |
| Air viscosity | $\mu$ (Pa·s) | 0.00001756 | 0.00001307 |
| Air density | $\rho$ (kg·m⁻³) | 0.00804 | 0.01161 |
| Thermal conductivity | $k$ (W·m⁻¹·K⁻¹) | 0.025367 | 0.018869 |
| Heat capacity | $C_p$ (J·kg⁻¹·K⁻¹) | 1039.6 | 1039.1 |
| Reynolds number | $Re$ | 2.29 | 4.44 |
| Heat transfer coefficient | $h$ (W·m⁻²·K⁻¹) | 66.5 | 54.4 |
| Solar irradiance | $S$ (W·m⁻²) | 1360 | 1360 |
| Absorptivity | $\alpha$ | 0.2 | 0.2 |
| **Radiation correction** | $T_s$ - $T_a$ (K) | **1.02** | **1.25** |

Although there are many parameters in Eq. (2), the decrease of thermal conductivity ($k$) of air dominantly causes the decrease of $h$ at low temperature. The thermal conductivity of air plays a main role in the heat transfer from sensor to air at the very boundary between them.

Since low temperature effect on radiation correction is due to a temperature-dependent change of material properties of air, the use of a low temperature chamber such as UAS is not the issue here. In this regard, the experimental observation of the low-temperature effect by the UAS is highly likely to be experienced by radiosondes during sounding.

In my opinion, the GRUAN data processing (GDP) of Vaisala RS41 should incorporate the effect of low temperature on the radiation correction because it is explained by both theoretical and experimental works.

The UAS paper (amt-2021-246) will be revised in the context of this comment soon. One simple way to include the low temperature effect is to use the Eq. (6) in the UAS paper. This may be possible when the GRUAN and the UAS papers are published back to back in the same Issue of the Journal with the help from the Editor.

---

## Author Response (AR2)

**Response to the Editor on the comments on the revised version of the manuscript AMT-2021-187**

We want to thank the Editor, Referee #1 and Kim Young-Gyoo (Referee #2) for their constructive comments and suggestions.

5

Editor:

Technical corrections:

10 "Eq. (1): Units are missing. Please specfiy the units for v, f, and p."

Units added to the sentence introducing Eq. (1), and in addition after Eq. (5).

"Table 3: "Sonde rotation" should be "Sonde rotation period"."

Changed to 'Rotation period'

15

"Eq. (5) and Figs. 13/14: The authors motivate their polynomial fit based on the inverse square root of both p and v.

In order to have a better link to the fit shown in Figs. 13/14, I propose to change v axis (currently linear) and p axis (currently log) into nonlinear axes which are linear in terms of
20 $1/\sqrt{p}$ and $1/\sqrt{v}$."

The main purpose of Figs 13 and 14 is to show the non-linear dependence of the temperature effect on the parameters p and v. Following the editor's recommendation, we have added in the Figures a version of the plots with the abscissa representing the reciproke of the square root of p or v. These additional plots show that the temperature
25 effect cannot solely be described by a linear dependence on $1/\sqrt{p}$ (or $1/\sqrt{v}$), but that additional polynomial terms are necessary, resulting in Eq. 5.

The caption of Fig. 14 is modified accordingly, and a short paragraph referring to the new panels in Fig. 14 is added at the end of Section 4.2.

30 "Figures: Copernicus now demands that color schemes are used which are clear for readers with colour vision deficiencies, see

https://www.atmospheric-measurement-techniques.net/submission.html#figurestables

Several of the color schemes used in this study do not fulfill this criterion; in particular the simultaneous use of green and red is problematic.

35 Please modify; you might also use different linestyles/markers for clarity."

Figures 3, 11, 14, 15, 16, 18 are adjusted to correspond to the guidelines. In Figs. 14 and 15 also various line styles are used.

40

Figure 19 has been tested with the Coblis color blindness simulator tool. For all the various filter settings it remained clear und unambiguous what the plot was displaying. Therefore, we have left the plot in this figure unchanged.

Referee #1:

Line 328: Better: "Here t is the time since opening the shutter and c0, c1, and c2 are fit parameters, where c0 represents the new equilibrium temperature, which is represented by

45 the horizontal green line in Fig. 11.

Legend for Figure 13: Delete "exemplary"

Legend for Figure 13: " … an overview over the magnitude and the distribution …"

Line 398: Better: "Of these four components, the uncertainty of ventilation speed is contributing most to the combined uncertainty u(DT)."

50 Line 401: Change the last word "that" to "which" preceded by a comma.

Line 403: Change "well" to "easily"

Line 404: Change "on" to "onto"

Lines 404f: Delete "beforehand defined"

Line 405: Change "linear" to "linearly"

55 Line 407: Remove the round brackets around "21x21" and add a comma in front of "which"

Line 408: Change "creation of" to "creating", change "are" to "is"

Line 430: Better "The Streamer model was used in the GRUAN processing of the Vaisala RS92 radiosonde data to estimate the radiation fluxes …"

Line 431: Better: "… in the data processing of the RS41 radiosonde data …"

60 Line 435: Better: "… the use of actual the measured position, pressure, temperature and humidity data and the use of representative values …"

Line 438: Better: "… of low solar elevation angles to improve the handling of the Earth's curvature …"

Line 554: Remove the round brackets around "2 to 3"

65 Line 588: Remove "then"

Lines 648f: Better: "of 3 m s-1 to 7 m s-1."

Line 675: Better: "… it does not have the option to cool the radiosonde …"

Line 682: Better: "…low temperatures. However, these experiments …""

70

All linguistic changes are included in the text as proposed.

Through personal communication and through the Referee comments from Dr. Yong-Gyoo Kim at KRISS on the revised version of our manuscript, we have learned that recent findings by the Korea Research Institute of Standard and Science (KRISS) confirm an increase of the radiative sensor heating at low temperature (−67 °C) and pressure (<30 hPa), which is mainly caused by the reduction of the convective heat transfer by air at low temperatures. We were not able to measure this effect in our setup, since SISTER is operated at room temperature.

Recently, colleagues from KRISS submitted a manuscript to AMT (Lee et al., AMT-2021-246), reporting on the increased sensor heating by solar radiation at low temperatures. However, since these new findings only become known to us during the second review of our manuscript, we prefer not to include a thorough discussion of this effect in our manuscript. Firstly, this would mean, in our estimate, a major revision of our manuscript, and secondly because the manuscript AMT-2021-246 still is under review. Furthermore, a major part of our manuscript (AMT-2021-187) is devoted to the description of the current correction algorithm in the GRUAN data processing for the RS41. If corroborated and published, the low temperature effect on the radiative sensor heating described in AMT-2021-246 will be accounted for in the next version of the GRUAN data processing for the RS41.

75

80

85

90